# Fossil fuel combustion and biomass burning sources of global black carbon from GEOS-Chem simulation and carbon isotope measurements

Ling Qi[1] and Shuxiao Wang[1,2]

[1] State Key Joint Laboratory of Environment Simulation and Pollution Control, School of Environment, Tsinghua University, Beijing 100084, China

[2] State Environmental Protection Key Laboratory of Sources and Control of Air Pollution Complex, Beijing 100084, China

*Correspondence to*: Shuxiao Wang (shxwang@tsinghua.edu.cn)

**Abstract.** We identify sources (fossil fuel combustion versus biomass burning) of black carbon (BC) in the atmosphere and in deposition using a global 3D chemical transport model GEOS-Chem. We validate the simulated sources against carbon isotope measurements of BC around the globe and find that the model reproduces mean biomass burning contribution ($f_{bb}$, %) in various regions within a factor of 2 (except in Europe, where $f_{bb}$ is underestimated by 63%). GEOS-Chem shows that contribution from biomass burning in the Northern Hemisphere ($f_{bb}$: 35±14%) is much less than that in the Southern Hemisphere (50±11%). The largest atmospheric $f_{bb}$ is in Africa (64±20%). Comparable contributions from biomass burning and fossil fuel combustion are found in South (S.) Asia (53±10%), Southeast (SE.) Asia (53±11%), S. America (47±14%), S. Pacific (47±7%), Australia (53±14%) and the Antarctic (51±2%). $f_{bb}$ is relatively small in East Asia (40±13%), Siberia (35±8%), the Arctic (33±6%), Canada (31±7%), the US (25±4%), and Europe (19±7%). Both observations and model results suggest that atmospheric $f_{bb}$ is higher in summer (59–78%, vary with sub-regions) than in winter (28–32%) in the Arctic, while it is higher in winter (42–58%) and lower in summer (16–42%) over the Himalayan–Tibetan plateau. The seasonal variations of Atmospheric $f_{bb}$ are relatively flat in North America, Europe, and Asia. We conducted four experiments to investigate the uncertainties associated with biofuel emissions, hygroscopicity of BC in fresh emissions, aging rate and size-resolved wet scavenging. We find that double biofuel emissions for domestic heating north of 45°N increases $f_{bb}$ values in Europe in winter by ~30%, reducing the discrepancy between observed and modeled atmospheric $f_{bb}$ from -63% to -54%. The remaining large negative discrepancy between model and observations suggests that the biofuel emissions are probably still underestimated at high latitudes. Increasing fraction of thickly coated hydrophilic BC from 20% to 70% in fresh biomass burning plumes increases the fraction of hydrophilic BC in biomass burning plumes by 0–20% (vary with seasons and regions), and thereby reduces atmospheric $f_{bb}$ by up to 11%. Faster aging (4 hour *e*-folding time versus 1.15 days of *e*-folding time) of BC in biomass burning plumes reduces atmospheric $f_{bb}$ by 7% (1– 14%, vary with seasons and regions), with the largest reduction in remote regions, such as the Arctic, the Antarctic and S. Pacific. Using size resolved scavenging accelerates scavenging of BC particles in both fossil fuel and biomass burning plumes, with a faster scavenging of BC in fossil fuel plumes. Thus, atmospheric $f_{bb}$ increases in most regions by 1–14%. Overall, atmospheric $f_{bb}$ is determined by $f_{bb}$ in

emissions mainly and by atmospheric processes, such as aging and scavenging, to a less extent. This confirms the assumption that $f_{bb}$ in local emissions determines atmospheric $f_{bb}$ in previous studies, which compared measured atmospheric $f_{bb}$ directly with local $f_{bb}$ in bottom-up emission inventories.

## 1 Introduction

Black carbon (BC) in the atmosphere and deposited over snow and ice absorbs solar radiation, triggers positive feedbacks and exerts a positive radiative forcing on the global climate (IPCC, 2014). Estimates of BC radiative forcing span a large range (0.2–1 W m$^{-2}$, Bond et al., 2013; IPCC, 2014). One of the uncertainties lies in the orders of magnitude different predictions of BC vertical profiles around the globe, particularly in remote regions, by chemical transport and climate models (Samset et al., 2013; 2014). To reduce the uncertainty, in addition to the widely used BC concentration observations

in the troposphere, at surface and in snow, observation-based source apportionment (fossil fuel versus biomass burning) of BC provides another dimension to constrain model simulations of BC distribution. The optical properties of BC from fossil fuel and biomass burning plumes are distinctively different (Bond et al., 2013), resulting in different radiative forcing from the two sources (Jacobson, 2010). Because of the relative short lifetime compared to greenhouse gases, accurate source apportionment of BC is important for short-term climate change mitigation.

Carbon isotope analysis is effective in distinguishing emissions from fossil fuel combustion (e.g. coal, oil and natural gas) and contemporary biomass burning (expressed as contribution from biomass burning, $f_{bb}$, %), because fossil emissions are $^{14}$C free and biomass emissions have a characteristic $^{14}$C/$^{12}$C ratio that is proportional to atmospheric carbon dioxide at the time of carbon fixation (Reddy et al., 2002). Combining $\delta^{13}$C and $\Delta^{14}$C measurements further differentiate the contribution from coal and liquid fossil fuel combustion (oil, gasoline and diesel, Andersson et al., 2015 and references therein). Fossil

fuel combustion has an anthropogenic origin, including industrial use, domestic cooking and heating, and transport (Bond et al., 2007). Contemporary biomass burning can come from both anthropogenic and natural sources. The former includes mainly industrial and domestic burning of biofuels (fuelwood, charcoal, agricultural residues, and dung, Fernandes et al., 2007) and the latter involves open fires of forests, crops, grass, and peatlands (van der Werf et al., 2010). Carbon isotope measurements are widely used for source apportionment of BC in the atmosphere in South Asia (Gustafsson et al., 2009;

Budhavant et al., 2015), East Asia (Chen et al., 2013; Andersson et al., 2015; Zhang et al., 2015; Li et al., 2016), Europe (Szidat et al., 2006; 2009; Zhang et al., 2012) and the Arctic (Barrett et al., 2015; Winiger et al., 2015; 2016; 2017), in snow over the Himalayan-Tibetan Plateau (Li et al., 2016) and in an Alpine ice core (Jenk et al., 2006).

Previous studies (Gustafsson et al., 2009; Chen et al., 2013; Li et al., 2016) compared carbon isotope measurements directly to $f_{bb}$ of local bottom-up emission inventories. The assumption behind these studies is that the major controlling factor of $f_{bb}$

in the atmosphere is local emissions. However, BC-containing particles in fossil fuel and biomass burning plumes have distinctively different mixing states and hygroscopicities (Moteki et al., 2007; Schwarz et al., 2008; Shiraiwa et al., 2007; Akagi et al., 2012), which might further affect BC scavenging in the two kinds of plumes and thus $f_{bb}$ in the atmosphere and

after deposition. Li et al. (2016) found smaller contribution from fossil fuel in snow than in air, suggesting that biomass burning emissions are easier to deposit compared to fossil fuel combustion emissions. Possible factors affecting $f_{bb}$ in the atmosphere and in deposition are mixing states and hygroscopicities in freshly emitted fossil fuel and biomass burning plumes, the consecutive aging rate and scavenging. However, as far as we are aware, no study has quantified the contribution of different factors to sources in terms of global BC in the atmosphere and in deposition.

In this study, we simulate sources of BC (fossil fuel combustion versus biomass burning) using a global 3D chemical transport model GEOS-Chem. We describe the model and the carbon isotope measurements in Sections 2 and 3, respectively. We evaluate the model simulation of $f_{bb}$ in Section 4.1, analyze the spatial and temporal variations of $f_{bb}$ in Section 4.2, evaluate the uncertainties associated with $f_{bb}$ in BC emissions, BC mixing state and hycroscopicity in fresh emissions, aging rage and size-resolved scavenging in Section 4.3.

## 2 Model description

GEOS-Chem is a global chemical transport model driven with assimilated meteorological fields from the Goddard Earth Observing System (GEOS) of the NASA Global Modeling and Assimilation Office (Bey et al., 2001). We use GEOS-Chem v11.01 coupled with the TwO Moment Aerosol Section (TOMAS) microphysics scheme (Adams and Seinfeld, 2002). This is a state-of-the-art global model to simulate global distribution of BC (Wang et al., 2011; Qi et al., 2017 (a and c)). We use 15 size bins ranging from 3 nm to 10 μm with tracers for sulphate, sea salt, organic aerosols, BC, and dust (Pierce et al., 2007; Lee et al., 2009; D'Andrea et al., 2013; Kodros and Peirce, 2017). Modern-Era Retrospective analysis for Research and Applications, Version 2 (MERRA2) meterological data set are used to drive model simulation at 4° latitude × 5° longitude horizontal resolution and 47 vertical layers from the surface to 0.01 hPa. Global fossil fuel and biofuel combustion emissions of BC are from Bond et al. (2007) and Fernandes et al. (2007), respectively. We also include gas flaring emissions from Stohl et al. (2013). We replace BC emissions in Asia by Li et al. (2017). We apply seasonal variations for domestic heating emissions based on degree-day concept (Stohl et al., 2013; Qi et al., 2017c). We use daily open fire emissions from Global Fire Emissions Database version 4 (GFED4, Giglio et al., 2013) in this study. We assume 20% of the freshly emitted BC aerosols are thickly coated and are hydrophilic (Park et al., 2003). We assume hydrophobic BC is converted to hydrophilic with an *e*-folding time of 1.15 days (Park et al., 2005). Wet deposition follows Liu et al. (2001), with updates of below cloud scavenging efficiency and in-cloud scavenging in ice clouds in Wang et al. (2011) and updates of BC scavenging in mix-phase clouds in Qi et al. (2017a).

## 3 Observation data

Carbon isotope analysis of BC sources in the atmosphere is available at 65 sites across the globe in different seasons to our knowledge (Table S1 and Fig. S3). Generally, $f_{bb}$ values are larger in remote regions (36±16% in South Asia, 33±14% in the

Arctic and 39±17% over the Himalayan–Tibetan plateau) than those in urban regions (13±4% in North America), indicating a larger contribution from biofuel and open fires in rural, developing and remote regions. In addition, $f_{bb}$ values strongly depend on seasons (see detailed analysis in Sect. 4.2.1). Carbon isotope measurements of BC in snow are only available over the Tibetan Plateau from Li et al. (2016).

5 Isotope mass balance equation based on the $\Delta^{14}C$ ($^{14}C/^{12}C$) data was applied to apportion the relative contributions to atmospheric BC from biomass burning of modern carbon ($f_{bb}$) and fossil fuel combustion.

$$\Delta^{14}C = \Delta^{14}C_{bb}\,f_{bb} + \Delta^{14}C_{ff}\,(1 - f_{bb}) \qquad (1)$$

10 Where $\Delta^{14}C$ is the measured radiocarbon content of the BC component and $\Delta^{14}C_{ff}$ is -1,000‰ by definition because fossil carbon is completely depleted in radiocarbon (Li et al., 2016). $\Delta^{14}C_{bb}$ end members used in this equation are usually between +70‰ and +225‰, depending on the type and age of the burned biomass (Winiger et al., 2015; Barrett et al., 2015; Li et al, 2016). The former value corresponds to freshly produced biomass, such as crop and grass. The latter value reflects the burn of wood, which has accumulated over the decades-to-century-long life span. Different choice of the $\Delta^{14}C_{bb}$ end member is 15 one of the uncertainties associated with this source apportionment method. Uncertainty of ±25‰ translates to < 5% in the resulting $f_{bb}$ estimate (Winiger et al., 2016). Another uncertainty stems from the method of isolating BC from total carbon in sampled particles (Zhang et al., 2012). They found that the isolation method prior to thermal treatments, thermal-optical methods, and the heating protocols are important to the isolation of BC and organic carbon and the following isotope analysis. They found that different protocols of thermal-optical method lead to ~ 30% difference of estimated $f_{bb}$ values.

20 **4 Results and Discussions**

GEOS-Chem captures the probability density function (PDF) of annual BC concentrations at sites in the US, Europe, China and the Arctic (see site description in Qi et al., 2017(b)) but overestimates the frequency of low BC concentrations (Fig. 1S (a)). About 30% of the simulated annual BC concentration in air is underestimated by a factor of 2 (Fig. 1S (b)). The model reproduces the PDF of BC concentration in snow preferably (correlation coefficient $r$ = 0.98, Fig. 2S (a)). The simulated 25 median BC concentrations in snow in various regions agree with observations within a factor of 2, except in region NC_Northeast Border (Fig. 2S (b)), where the model overestimates the observed BC concentration in snow by a factor of 3 due to the overestimate of local emissions in that region (Qi et al., 2017b).

**4.1 Contribution of biomass burning to BC in various regions**

GEOS-Chem simulated mean atmospheric $f_{bb}$ in each region agrees with observations within a factor of 2, except in Europe, 30 where $f_{bb}$ is underestimated by 63% (Fig.1 (a)). The low bias of $f_{bb}$ in Europe occurs in non-summer seasons (observation: 45%, model: 13%), which is partly due to the underestimate of biofuel combustion for domestic heating by Fernandes et al.

(2007) in most of the European regions during cold seasons (Herich et al., 2011). In South (S.) Asia, mean atmospheric $f_{bb}$ is overestimated by 50%, mostly from the 90% overestimate of $f_{bb}$ at Delhi (observation: 28%, model: 52%). At this site, atmospheric $f_{bb}$ in spring and summer are overestimated by 100% and 200%, respectively. In North America, the model overestimates $f_{bb}$ at Salt Lake City (SLC) and Mexican City by a factor of 2. Possible reasons for the overestimate are explained in Sect. 4.2.1. In the Arctic and East (E.) Asia, the model reproduces the observed $f_{bb}$ values within 3% and 7%, respectively. In addition, GEOS-Chem underestimates the large variations of $f_{bb}$ values (horizontal lines in Fig. 1 (a)) in every region (except in the Arctic), due to the coarse horizontal and vertical resolutions.

Over the Himalayan–Tibetan plateau, observations show that biomass burning dominates BC deposited in snow (64%), but its contribution in the atmosphere is much less (39%, Li et al., 2016). GEOS-Chem reproduces the average $f_{bb}$ in snow (model: 63%) but overpredicts the average atmospheric $f_{bb}$ (model: 62%) by 56%. GEOS-Chem simulated $f_{bb}$ values of BC deposition in snow at all sites over the Himalayan–Tibetan plateau agree with observations within 40% during both monsoon (June–August) and non-monsoon seasons (Fig. 1 (b)), suggesting that the model captures the spatial and temporal variations of $f_{bb}$ in BC deposition in this region. The overestimate of the atmospheric $f_{bb}$ is mainly from 130% overestimate of $f_{bb}$ during monsoon season (observation: 29%, model: 67%). Possible reasons for the overestimate are discussed in Sect. 4.2.1.

## 4.2 Temporal and spatial variations of $f_{bb}$ in different regions

### 4.2.1 Temporal variation of $f_{bb}$

In the Arctic at Abisko, observed $f_{bb}$ ranges from fall and wintertime low of 31% to summer high of 59% (Fig. 2(a)), due to the large contribution from open fires in Europe in summer (Winiger et al., 2016). The model also shows a peak of $f_{bb}$ in summer, but the seasonal variation is relatively flat (from 23% in winter to 27% in summer). We attribute the discrepancy to two reasons. First, $f_{bb}$ values of emissions at the site lack seasonal variations, as shown in Fig.2(a). Second, the coarse resolution does not solve the vortex structure of the low-pressure and frontal systems, which is important for poleward transport of BC (Ma et al., 2014; Sato et al., 2016). At Barrow (Fig. 2 (b)), observed $f_{bb}$ show two peaks in summer (34%) and winter (37%), while modeled $f_{bb}$ shows a single strong peak in summer (78%). In summer, the magnitude and variations of $f_{bb}$ in the atmosphere is similar to that of $f_{bb}$ in local emissions, suggesting that the atmospheric $f_{bb}$ is largely determined by local emissions. The 129% overestimate of $f_{bb}$ is largely due to the overestimate of local open burning emissions. In spring, fall and winter, the modeled atmospheric $f_{bb}$ values are much larger than the $f_{bb}$ of local emissions, indicating a large contribution from long-range transport.

In contrast to the seasonal cycles of $f_{bb}$ at sites in the Arctic, at Bode (Fig. 2(c)) over the Himalayan–Tibetan Plateau, $f_{bb}$ values are the lowest in summer (observation: 17%) and highest in winter (observation: 42%, Li et al., 2016). Similar trend is observed at Lumbini (Fig. 2(d)), only with smaller amplitude (summer low: 42%, spring high: 58%, Li et al., 2016). The lower $f_{bb}$ in summer is because of several reasons. First, less biofuel is consumed for domestic heating in warmer seasons (Li et al., 2016). Second, the region is barely affected by open fires. Third, biomass-sourced BC is removed more efficiently by

the frequent precipitation in summer both over the Himalayan–Tibetan plateau and over the surrounding source regions, such as India and East Asia (Li et al., 2016). The GEOS-Chem simulated atmospheric $f_{bb}$ of BC at all sites over the Himalayan–Tibetan plateau (results for Bode and Lumbini are shown in Fig. 2 (c) and (d) and the others are not) have weak or no seasonal variations. In addition, the model does not capture the observed increasing trend of $f_{bb}$ along the Mustang valley and Langtang valley. Possible reasons for the discrepancies are several folds. First, the $f_{bb}$ values of local emissions have no seasonal variations, as shown in Fig. 2 (c) and (d). Second, it is conceivable that the coarse model resolution of global models does not reproduce the complex topography and transport pathways of BC over the Himalayan–Tibetan plateau (He et al., 2014). However, the mean modeled atmospheric $f_{bb}$ generally agrees with observations (within 60%) and the modeled atmospheric $f_{bb}$ generally follows the $f_{bb}$ of local emissions across the whole plateau. These comparisons suggest that the atmospheric $f_{bb}$ over the Himalayan–Tibetan plateau is largely determined by $f_{bb}$ in emissions in the region.

At sites SLC (North America, Fig. 2(e)), Tokyo (East Asia, Fig. 2(f)), MCOH and SINH (South Asia, Fig. 2(g) and (h)), no big differences of $f_{bb}$ among seasons were observed (SLC: 8–13%; Tokyo: 33–41%; MCOH: 52–53%; SINH: 48–56%). However, BC concentrations show strong seasonal variations at the four sites, with high loadings in winter and low loadings in summer (Mouteva et al., 2017; Yamamoto et al., 2007; Budhavant et al., 2015). At SLC, the most significant local sources of PM$_{2.5}$ particles are mobile emissions, which are relatively stable through the whole year (Mouteva et al., 2017). The second most important source is non-mobile sources with solid burning, mostly wood burning, which is not allowed to use when air quality forecasts predict an inversion period (Mouteva et al., 2017). This restriction limits the extra use of solid fuels in winter, and thus limited their effects on BC concentrations and $f_{bb}$ in the atmosphere. So the higher concentration of BC in winter in SLC is largely determined by the low boundary layer height (Mouteva et al., 2017). The model overestimates $f_{bb}$ at SLC in all seasons by a factor of 2–4 (Fig. 2(e)). As described in Mouteva et al. (2017), the observations were in urban environment with strong influence from local emissions. However, modeled $f_{bb}$ in the atmosphere is much higher than the $f_{bb}$ values of local emissions based on emission inventories in this study (Sect. 2), suggesting that the modeled atmospheric $f_{bb}$ at the site is largely affected by the surrounding regions. The misrepresentation of source region (local versus regional) is probably one reason of the large bias of modeled $f_{bb}$ against observations. At site Tokyo, East Asia, the model reproduces both the magnitude and the seasonal variations of observed $f_{bb}$. The much lower $f_{bb}$ value in emissions than in the atmosphere also indicates a regional effect. In South Asia, GEOS-Chem reproduces the similar observed high $f_{bb}$ values at MOCH (summer: 52%; winter: 53%) and SINH (summer: 48; winter: 56%) within 30%. However, reasons for the high $f_{bb}$ values at the two sites are different. Since there are no local emissions at MCOH, $f_{bb}$ at the site is largely affected by long-range transport. In contrast, $f_{bb}$ in the atmosphere follows $f_{bb}$ in local emissions at SINH, suggesting that the atmospheric $f_{bb}$ at the site is mostly affected by local emissions. At MCOH the high $f_{bb}$ is probably from the large $f_{bb}$ in the outflow of Africa, while at SINH local burning of agricultural crop residues are the major sources (Budhavant et al., 2015).

### 4.2.2 Spatial variation of modeled $f_{bb}$

GEOS-Chem suggests that the Southern Hemisphere has a higher contribution from biomass burning both for BC in surface air (50±11%) and in deposition (53±10%, Fig. 3 (a) and (b)). The high $f_{bb}$ in S. America and Australia are largely from active open fires (accounting for 48% and 81% of the total biomass burning contributions, respectively), while in Africa biofuel consumption is the major biomass burning source (model: 64±20%, Fig. 3 (c) and (d)). Because of the strong seasonal variations of open fire emissions, the highest $f_{bb}$ in Africa, S. America, S. Pacific, Australia and the Antarctic usually occur during September to November (58–71%), and the lowest values are in March–May (32–56%, Fig. S4).

In the Northern Hemisphere, the largest $f_{bb}$ of both BC in the atmosphere (93±5%) and in deposition (92±6%) are in North Congo, where biomass burning contribution dominates over fossil fuel emissions. South Asia also shows large $f_{bb}$ (54% for BC in air and in deposition) due to large biofuel consumption. In other regions, such as Europe, Canada, the US, Siberia and the Arctic, fossil fuel contribution (65–80%) is much larger than biomass burning. $f_{bb}$ of BC in air and in deposition in different regions have different seasonal variations (Figs. S4–S5). Atmospheric $f_{bb}$ in Canada, Siberia, the Arctic and the Antarctic have the strongest seasonal variations with a peak in summer (49–70%) because of the large fraction of open fire emissions (Fig. S6–S7). In the US, South Europe, East Asia and South Asia, seasonal variation of $f_{bb}$ is relatively flat, which is also shown by observations at a few sites (Fig. 2).

### 4.3 Uncertainty analysis

Atmospheric $f_{bb}$ is determined not only by emissions (fossil fuel combustion versus biomass burning), but also by atmospheric processes that affect the deposition during transport. We investigate the uncertainties associated with biofuel emissions, $f_{bb}$ in fresh emissions, BC aging rate and size-resolved scavenging. We used relative change ($r$, %) to describe the change of $f_{bb}$ in each experiment (Exp.) relative to the standard simulation.

$$r = ([f_{bb}]_{Exp.} - [f_{bb}]_{Std.})/[f_{bb}]_{Std} \qquad (2)$$

where $r$ is the relative change, $[f_{bb}]_{Exp.}$ is $f_{bb}$ in each experiment and $[f_{bb}]_{Std}$ is the $f_{bb}$ in the standard simulation in each region.

### 4.3.1 Uncertainty associated with biofuel emissions

Biofuel emission estimates are associated with large uncertainties (Fernandes et al., 2007). Source apportionment of BC in Europe based on multi-wavelength aethalometer measurements showed that $f_{bb}$ in winter (24–33%) is much higher than that in summer (2–10%), suggesting that wood burning for domestic heating increases the $f_{bb}$ value in the atmosphere in winter significantly (Herich et al., 2011). In addition, Winiger et al. (2017) analyzed $f_{bb}$ based on carbon isotope measurements at Tiksi in Russia and suggested that domestic (~60% of which is from biomass burning) accounted for 35% of BC at the site,

following transport (38%). We find that during cold season mean $f_{bb}$ values in Europe and the Arctic (most sites are north of 45°N, Table S1) are underestimated by 68% and 50% in the standard simulation, probably due to the underestimate of domestic heating in winter. However, in East Asia (all sites are south of 45°N), mean $f_{bb}$ in winter is overestimated by 22%. Thus, we doubled biofuel emissions from domestic heating north of 45°N during cold seasons in Experiment (Exp.) A to

investigate the uncertainty associated with biofuel emissions. It is conceivable that the largest effects occur in the Northern four regions, including Europe, Siberia, Canada and the Arctic. As a result, $f_{bb}$ values increase by ~30% in Europe, Siberia and the Arctic and by 15% in Canada in winter, larger than that in spring and fall (4–13%, Fig. 4). Consequently, the low bias of $f_{bb}$ in Europe is reduced from -63% to -54%. This improvement suggests that the biofuel emissions at high latitudes in the Northern Hemisphere are probably too low in current bottom-up BC emission inventories, supporting previous estimates

(Herich et al., 2011).

**4.3.2 Uncertainty associated with hygroscopicity of BC in freshly emitted biomass burning plumes**

Recent measurements find that in freshly emitted fossil fuel plumes the fraction of thickly coated hydrophilic BC is ~10% (Moteki et al., 2007; Schwarz et al., 2008; Shiraiwa et al., 2007), while in biomass burning plumes the fraction reaches up to 70% (Schwarz et al., 2008; Akagi et al., 2012). The higher hygroscopicity of BC in freshly emitted biomass burning plumes

enhances the subsequent wet scavenging rate and thereby reduces $f_{bb}$ in the atmosphere. In the standard simulation, we assume 20% of freshly emitted BC particles are hydrophilic. We investigate the effects of the initial hygroscopicity of BC in fresh emissions on atmospheric $f_{bb}$ of BC in Exp. B by assuming 70% of freshly emitted BC particles from biomass burning are thickly coated and hydrophilic. The resulting fraction of hydrophilic BC in biomass burning plumes in the 12 regions increase by 0–20% (vary with seasons and regions), lowering $f_{bb}$ in the atmosphere by up to 11% in Canada in summer. The

largest reduction of $f_{bb}$ shows in June–August (-7% averaged for all regions, Fig. 4), when open fires are frequent and active globally (Giglio et al., 2013; van der Werf et al., 2010). During this time, the largest reductions are in Canada (-11%) and Siberia (-10%), where the fraction of hydrophilic BC in biomass burning plumes increases by a large fraction (11–13%). In S. Pacific, the reduction of $f_{bb}$ is large (-10%) as well, because large precipitation (28 kg m$^{-2}$ mon$^{-1}$) over this region removes more biomass burning BC particles in the outflow of S. America. During September–November, the relative reduction of $f_{bb}$

in the Northern Hemisphere (-6%) is much larger than that in the Southern Hemisphere (-1%), because $f_{bb}$ values in the Southern Hemisphere are too large (Fig. S5). The changes of $f_{bb}$ values in other seasons in all regions are marginal.

**4.3.3 Uncertainty associated with BC aging time**

Mixing with organic and inorganic particles with larger hygroscopicity, BC particles become more hydrophilic during aging process (Bond et al., 2013). It is assumed that BC particles are converted from hydrophobic to hydrophilic with an *e*-folding

time of 1.15 days after emission in the standard simulation (Park et al., 2005). However, observations showed that the fraction of thickly coated hydrophilic BC in urban fossil fuel plumes increases linearly with plume age (0.5–2.3% h$^{-1}$, Moteki et al., 2007; Shiraiwa et al., 2007; Subramanian et al., 2010; McMeeking et al., 2011), while BC aging follows a logarithmic

trend with an *e*-folding time of 4 hours in biomass burning plumes (Akagi et al., 2012). The aging rates differ among plumes because of different BC sizes, co-emitted hygroscopic materials and oxidation capacities of the plumes (Bond et al., 2013). Thus, in Exp. C, we assume fossil fuel combustion generated BC ages linearly with a rate of 1% h$^{-1}$, while BC from biomass burning plumes ages with an *e*-folding time of 4 hours. This means that the fossil fuel plumes age slower than the standard simulation and be scavenged slower, while the biomass burning plumes age much faster and are removed from the atmosphere faster in precipitation. This aging scheme leads to a 0–24% increase of fraction of hydrophilic BC in the atmosphere, which reduces $f_{bb}$ by up to -14%. The largest reduction of $f_{bb}$ is in S. Pacific in fall (MAM) and summer (DJF) in the Southern Hemisphere, followed by the Antarctic (-12%) during MAM and the Arctic (-11%) during SON. The reduction of $f_{bb}$ is larger in remote regions and smaller in source regions, because it takes time for the different aging rates in fossil fuel and biomass burning plumes to affect the hygroscopicities of BC in the two plumes and the subsequent aging rates.

### 4.3.4 Uncertainty associated with size resolved scavenging

BC particles emitted from biomass burning plumes are usually larger in size and thicker in coating thickness (Schwarz et al., 2008; Sahu et al., 2012), suggesting an easier removal from the atmosphere. For example, observations (Schwarz et al., 2008; Sahu et al., 2012) showed that the mass median diameter of BC particles in biomass burning plumes is 193 nm with a coating thickness of 65 nm, while in fossil fuel plumes, the mass median diameter and coating thickness are 175 nm and 20 nm. In addition, because of the different coating materials, hygroscopicities of BC-containing particles in the two kinds of plumes are different as well. The coating materials of BC in urban plumes are dominated by sulfate and followed by nitrate and primary and secondary organics (Shiraiwa et al., 2007), while in biomass burning plumes, the major coating materials are organics (Sahu et al., 2012). For ambient air, characteristic $\kappa$ values of organics and inorganics are 0.1 (0.01–0.5) and 0.7 (0.5–1.4, Petters and Kreidenweis, 2007; Gunthe et al., 2011 and references therein). Higher hygroscopicity of BC in fossil fuel plumes suggests that they are easier to be activated and serve as CCN compared to BC particles in biomass burning plumes. The higher hygroscopicity and smaller size of BC particles in fossil fuel plumes have opposite effect on their removal rate. Thus, we investigate the total effects of size-resolved scavenging in Exp. D. we use the TOMAS microphysics scheme to process the aging and wet scavenging of BC with different sizes from fossil fuel combustion and biomass burning. The mass median diameters of fossil fuel and biomass burning BC particles are assumed to be 160 nm and 200 nm, respectively. Size resolved coagulation, condensation, nucleation and cloud processing are implemented. Coating materials included are sulfate, nitrate, sea-salt, organics and mineral dust. The size-resolved aging and scavenging scheme leads to a larger increase of fraction of hydrophilic BC in fossil fuel plumes (by 16% (0–31%, vary with regions)) than in biomass burning plumes (by 12% (0–23%)). This increase in both fossil fuel and biomass burning plumes suggest that BC particles are removed faster in the size-resolved simulation than in the standard simulation with a bulk removal parameterization. The larger increase of the fraction of hydrophilic BC in fossil fuel plumes means that BC in fossil fuel plumes are removed faster than those in biomass burning plumes in the size-resolved simulation. This is probably because the total effect of higher hygroscopicity of coating materials and smaller size of BC in fossil fuel plumes is enhancing their removal. Thus

atmospheric $f_{bb}$ increases in most regions during MAM (by 1–14%), SON (by 0–7%) and DJF (by 1–12%). The most noticeable characteristics is that the increase of $f_{bb}$ in Northern Hemisphere is larger than those in Southern Hemisphere, due to the large fraction of fossil fuel emissions in the Northern Hemisphere.

### 4.3.5 Uncertainty associated with model resolution

Finer model resolution is capable to reproduce small-scale meteorological conditions, which is critical to BC transport (Sato et al., 2016). We use horizontal resolution of 4° lat × 5° lon in the standard simulation and Exps. A–D, because the size-resolved microphysical scheme TOMAS in Exp. D is computationally expensive. We investigate the uncertainty associated with model resolution in Exp. E by using a finer horizontal resolution of 2° lat × 2.5° lon (Fig. 4). We find that relative to the standard simulation, $f_{bb}$ in Exp. E changes by -5%–5% in the 13 regions in all seasons. In most regions, the absolute change

is smaller than or equal to the change in Exp. A–D, except in mid-latitude and tropical regions in Exp. A. Averaged in the whole globe, the relative change of $f_{bb}$ to the standard simulation is -1%.

### 4.3.6 Other uncertainties

Carbon isotope measurements of BC sources are associated with large uncertainties. Thermal-optical protocol used for the carbon isotope measurements of BC produce ~30% difference of observed $f_{bb}$ values (Zhang et al., 2012), which is equal or

larger than the uncertainties of modeled $f_{bb}$ associated with biofuel emissions north of 45°N, aging rate and wet scavenging discussed in Sect. 4.3.1-4.3.4. The comparison of the two sets of data in Sect. 4.1 and 4.2 are within similar uncertainty range. In addition, we do not have carbon isotope measurements in the Southern Hemisphere to constrain the model results. Our analysis in this study is based only on model results.

In addition to the biofuel emissions discussed in Sect 4.3.1, open fire emissions, particularly in the boreal regions, are

associated with large uncertainties (Randerson et al., 2012). Konovalov et al. (2018) found that open burning emissions of Siberian fires during May to September from GFED4 is possibly underestimated by a factor of 2 constrained by satellite observations of the aerosol absorption optical depth and the aerosol extinction optical depth. However, we find that during the same season, mean atmospheric $f_{bb}$ at Tiksi in Russia is overestimated by 88%, indicating that open burning emissions in this region from GFED4 are possibly overestimated. This contradiction suggests that further studies are needed to better

constrain the open burning emissions in boreal regions. In addition, the global fossil fuel (Bond et al., 2007) and biofuel emission inventory (Fernandes et al., 2007) used in this study are for year 2000 and the emissions in Asia (Li et al., 2017) are for year 2010. We estimated the $f_{bb}$ from 2007 to 2013 using these constant inventories and varying open burning emissions from GFED4. The lack of inter-annual variations of BC fossil fuel and biofuel emissions also produces uncertainties, but it is difficult to quantify based on current knowledge.

# 5 Conclusions

This study sought to understand the relative contribution of fossil fuel combustion and biomass burning to global BC. We used GEOS-Chem (version 11-01-01) driven by MERRA2 assimilated meteorological fields to simulate BC concentration from fossil fuel and biomass burning. The source apportionment results were expressed as the fraction of BC from biomass burning ($f_{bb}$). Simulated $f_{bb}$ was validated against carbon isotope measurements of BC in the atmosphere at 65 stations across the Northern Hemisphere and 11 snow samples over the Himalayan–Tibetan plateau. We also investigated the uncertainties of $f_{bb}$ associated with biofuel emissions, fraction of hydrophilic BC in fresh emissions, aging time and size-resolved scavenging.

The model reproduced the mean observed atmospheric $f_{bb}$ in various regions and in snow over the Himalayan–Tibetan plateau within a factor of 2. Generally, values of atmospheric $f_{bb}$ were larger in remote regions (33±14% in the Arctic, 39±17% over the Himalayan–Tibetan plateau and 36±16% in South Asia) than those in urban regions (13±4% in North America), indicating a larger contribution from biofuel and open burning sources in rural, developing and remote regions. $f_{bb}$ was higher in summer (59–78%, vary with regions) than in winter (28–32%, vary with regions)) in the Arctic, while it was higher in winter (42–58%, vary with regions)) and lower in summer (16–42%, vary with regions)) over the Himalayan–Tibetan plateau. The simulated amplitudes of the seasonal variations were much smaller in the two regions. The seasonal variation was observed to be relatively flat in North America, East and South Asia. The simulated monthly mean $f_{bb}$ in these regions agree with observations by -45–275%. The Southern Hemisphere had a higher atmospheric $f_{bb}$ than the Northern Hemisphere (SH: 50±11%, NH: 35±14%) due to the large fraction of open burning emissions in S. America and Australia and large fraction of biofuel consumption in Africa. In the Northern Hemisphere, the highest $f_{bb}$ was in S. Asia (54±10%), followed by E. Asia (41±13%), due to large biofuel consumption. In other regions, such as Europe, Canada, the US, Siberia and the Arctic, $f_{bb}$ values are small (20–35%, vary with regions)).

Simulated $f_{bb}$ was associated with uncertainties from all processes, including emissions, aging and deposition processes. We found that doubled biofuel emissions used for domestic heating north of 45°N resulted in a ~30% increase of $f_{bb}$ in Europe, Siberia and the Arctic and a 15% increase in Canada in winter. This increase reduced the discrepancy of $f_{bb}$ against observations from -63% to -54% in Europe, suggesting that the biofuel emissions at high latitudes were underestimated by the bottom-up emission inventories. Using a higher fraction of hydrophilic BC in fresh biomass burning plumes (uncertainty simulation: 70%, standard simulation: 20%) resulted in a reduction of $f_{bb}$ in summer by -2 – -11%, with the largest reduction in Canada and Siberia, where open fires were frequent. In the standard simulation, it was assumed that BC in both fossil fuel and biomass burning plumes aged following an *e*-folding time of 1.15 days. In the uncertainty simulation, we used a 4 hour *e*-folding life time for BC in biomass burning plumes and a linear aging rate of 1% for BC in fossil fuel plumes. This led to a reduction of $f_{bb}$ up to -14% in the atmosphere. The largest reduction was in S. Pacific in fall (MAM) and summer (DJF) in the Southern Hemisphere. The reductions in the Antarctic (-12%) and the Arctic (-11%) were also large in fall when there were large open fires in the Southern Hemisphere and at high latitudes in the Northern Hemisphere. Size-resolved aging and

scavenging scheme led to a larger increase of fraction of hydrophilic BC in fossil fuel plumes (by 16% (0–31%)) than in biomass burning plumes (by 12% (0–23%)). Thus atmospheric $f_{bb}$ increased in most regions during MAM (by 1–14%), SON (by 0–7%) and DJF (by 1–12%). Using finer model resolution produced -5%–5% relative change of atmospheric $f_{bb}$ in the various regions, equal or smaller than the change caused by atmospheric processes.

This study showed that local emissions had a larger effect on atmospheric $f_{bb}$ than other atmospheric processes. As discussed in Sect. 1, most previous studies compared measured atmospheric $f_{bb}$ directly with $f_{bb}$ in local emissions. We confirmed this assumption, but suggested considering the uncertainties associated with aging and scavenging (up to 14%). In addition, ~30% difference of isotope-based measurements of $f_{bb}$ caused by the thermal-optical protocols in measuring BC should also be considered.

This study has important implications for estimating radiative forcing of global BC. Previous studies (Healy et al., 2015 and references therein) showed that BC-containing particles in open fires had no optical lensing effect. Considering the large contribution from biomass burning in S. Asia, SE. Asia and in the Southern Hemisphere as suggested in this study, the inclusion of lensing-related absorption enhancement in climate models for BC from both fossil fuel combustion and biomass burning sources may lead to an overestimate of the radiative forcing of global BC. Measurements of the optical properties of

BC particles from different sources (fossil fuel versus biomass burning) in different regions are needed to better constrain its radiative forcing.

**Code/Data availability**

The data used in this study are available from the corresponding author upon request ([shxwang@tsinghua.edu.cn](mailto:shxwang@tsinghua.edu.cn)).

**Author contribution**

Ling Qi and Shuxiao Wang designed the experiments. Ling Qi performed the simulations. Ling Qi prepared the manuscript with contributions from Shuxiao Wang.

**Competing interests**

The authors declare that they have no conflict of interest.

**Acknowledgments**

This work was supported by Key Projects of National Key Research and Development Program of the Ministry of Science and Technology of China (2017YFC0213005), the National Natural Science Foundation of China (21625701 and

21806088), and National research program for key issues in air pollution control (DQGG0301 and DQGG0303). We thank the two reviewers for their constructive comments on the manuscript.

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

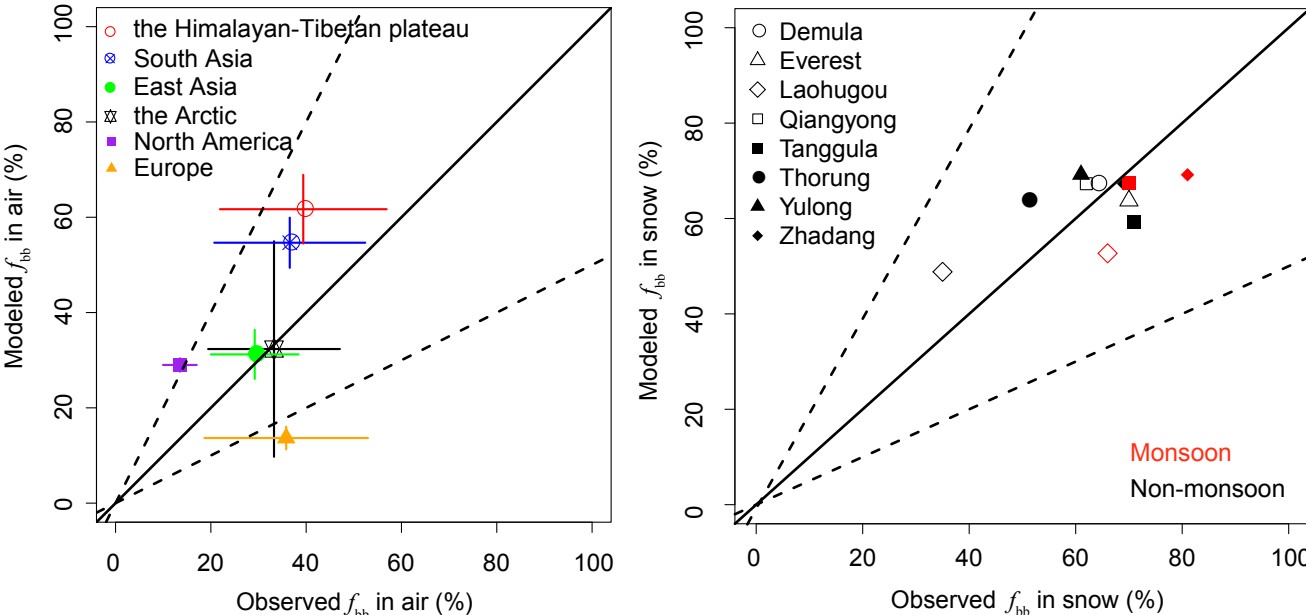

**Figure 1: Observed and GEOS-Chem simulated fraction of biomass burning ($f_{bb}$, %) of (a) BC in the atmosphere in the Arctic, South Asia, North America, Europe, East Asia, and the Himalayan-Tibetan plateau (the regions are symbol and color coded, see data in Table S2.) and (b) BC in snow during monsoon (red) and non-monsoon (black) seasons over the Himalayan-Tibetan plateau. Also shown in (a) are the standard deviations of observed and model simulated $f_{bb}$ in each region, reflecting the temporal and spatial variations of $f_{bb}$ in the region (horizontal and vertical lines). Observations of $f_{bb}$ in the atmosphere in (a) are from carbon isotope analysis as listed in Table S1. Observations of $f_{bb}$ in BC in snow in (b) are from Li et al. (2016). Solid lines in (a) and (b) are 1:1 ratio lines and dashed lines are 1:2 (or 2:1).**

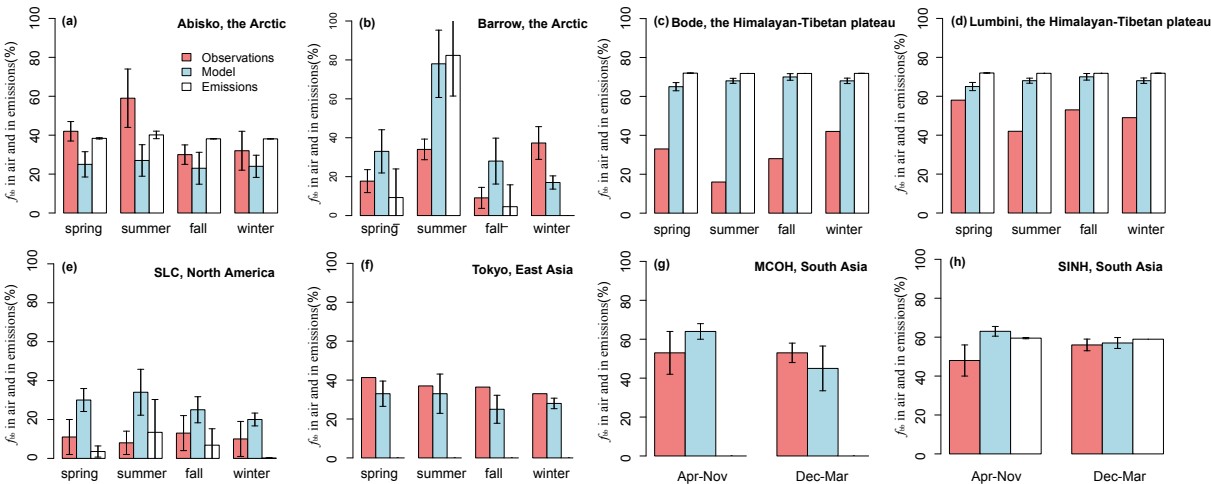

**Figure 2: Seasonal variations of observed (lightcoral bars) and GEOS-Chem simulated (lightblue bars) $f_{bb}$ of BC in the atmosphere at (a) Abisko and (b) Barrow in the Arctic, (c) Bode and (d) Lumbini over the Himalayan–Tibetan Plateau, (e) Salt Lake City in North America, (f) Tokyo in East Asia, (g) MCOH and (h) SINH in South Asia. The white bars are $f_{bb}$ values of BC emissions in the model grid (4° lat x 5° lon) of each site. Also shown are the standard deviations (error bars). Site locations are shown in Fig. S3.**

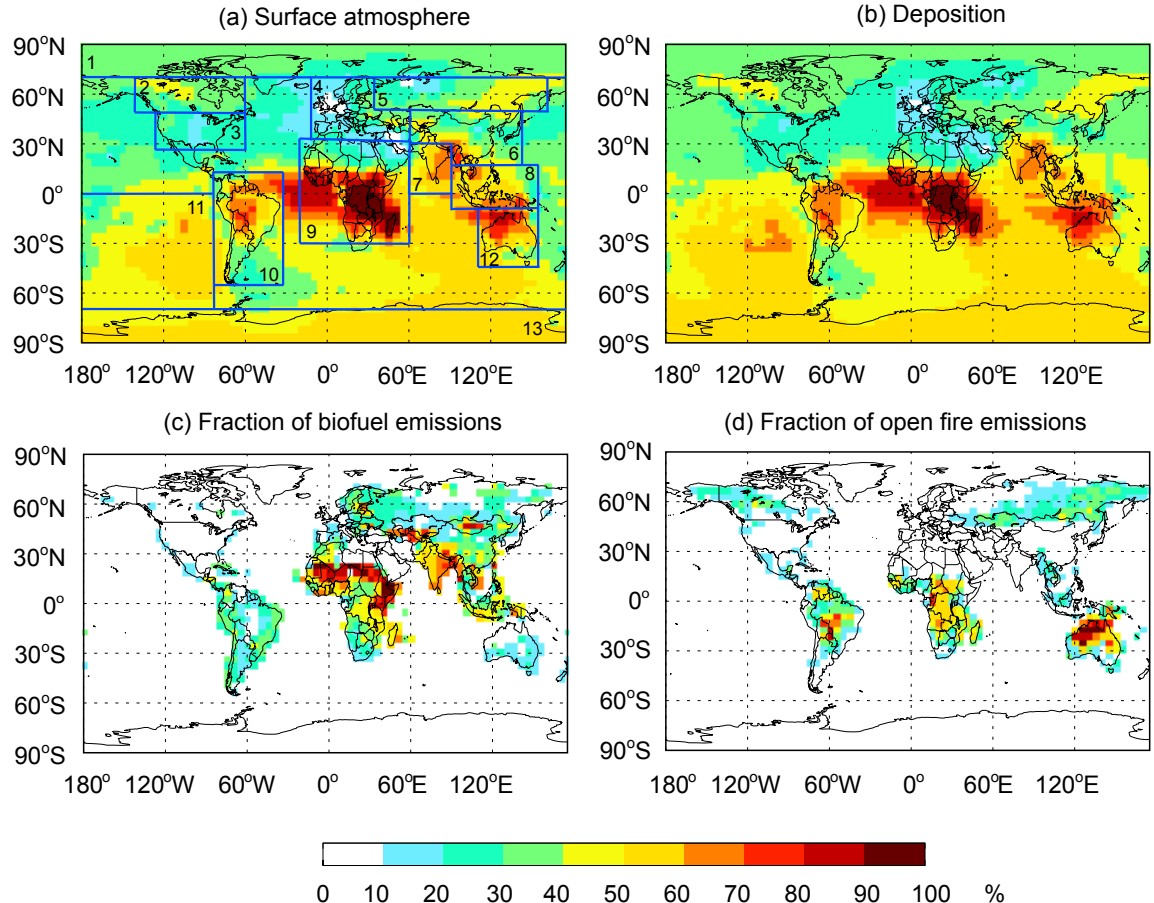

**Figure 3:** Annual (a) $f_{bb}$ of BC in the atmosphere at surface, (b) $f_{bb}$ of BC deposition, (c) fraction of biofuel emissions and (d) fraction of open fire emissions. Data are averaged for 2007–2013. Also shown in (a) are regions discussed in the text: 1. the Arctic, 2. Canada, 3. the US, 4. Europe, 5. Siberia, 6. East (E.) Asia, 7. South (S.) Asia, 8. Southeast (SE.) Asia, 9. Africa, 10. S. America, 11. S. Pacific, 12. Australia, and 13. the Antarctic.

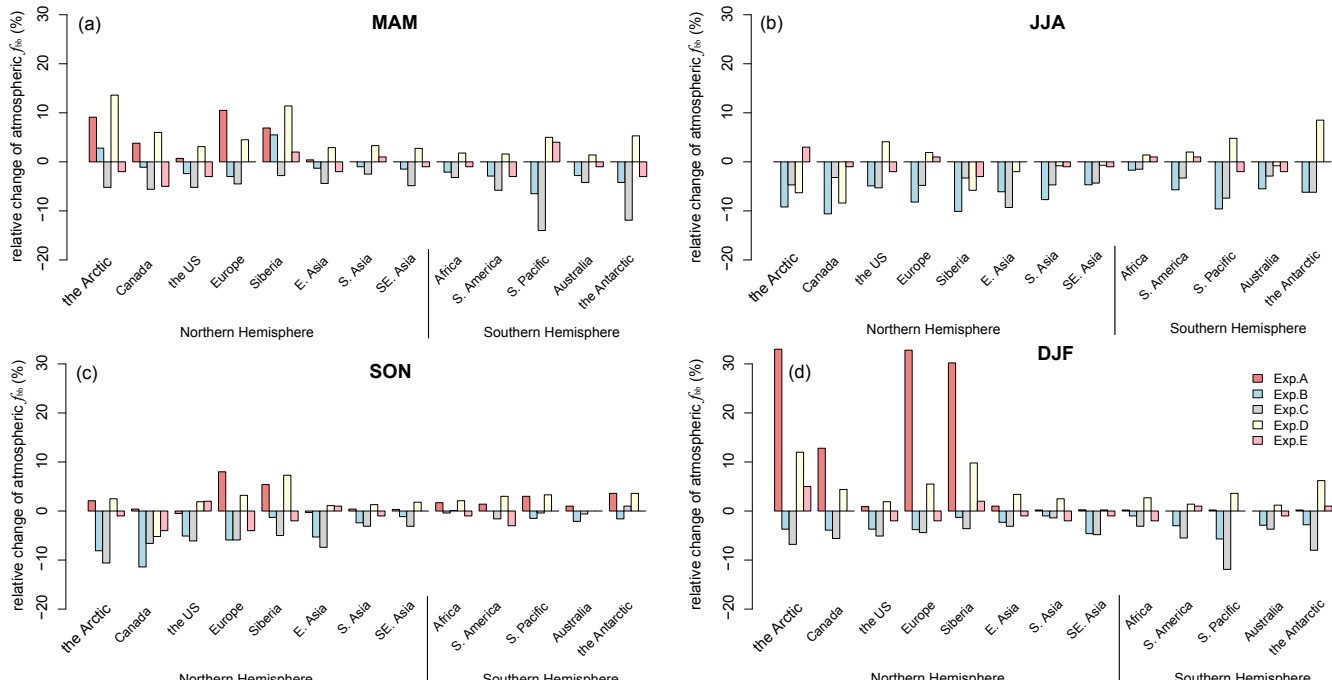

**Figure 4: GEOS-Chem simulated fractional change (*r*) to atmospheric *f*$_{bb}$ relative to the standard simulation, as a result of doubled biofuel emissions north of 45°N (Exp. A), 70% of hydrophilic BC in freshly emitted biomass burning BC-containing particles (Exp. B), 4 hour *e*-folding aging time of BC in biomass burning plumes and linear aging rate of 1% in fossil fuel plumes (Exp. C), TOMAS microphysical aging and scavenging (Exp. D) and finer horizontal model resolution (2° lat × 2.5° lon, Exp. E), *r* = ([*f*$_{bb}$]$_{Exp.}$ − [*f*$_{bb}$]$_{Std.}$)/[ *f*$_{bb}$]$_{Std}$, that varies with regions (see region definition in Fig.3 (a)) and seasons ((a) March–May (MAM), (b) June–August (JJA), (c) September–November (SON) and (d) December–February (DJF)), averaged for 2007–2013. See details of the standard simulation and the uncertainty experiments in the text.**