# Peer review of "Fossil fuel combustion and biomass burning sources of global black carbon from GEOS-Chem simulation and carbon isotope measurements"

_Atmospheric Chemistry and Physics, 2019_

## Referee Comment (RC1) · Anonymous Referee #2 · 1 May 2019

Referee for
Journal: ACP
Title: Fossil fuel combustion and biomass burning sources of global black carbon
Author(s): Ling Qi and Shuxiao Wang
MS No.: acp-2019-46

General comments

This manuscript represents an extensive approach to compare black carbon (BC) fractions of biomass burning from observations all across the northern hemisphere to GEOS-Chem simulations. The authors further conduct four sets of experiments to test uncertainties associated with the modelling of BC biofuel emissions, BC hygroscopicity, BC ageing, and BC size-resolved scavenging.

The overall presentation, language, and figures are of high quality. The study is within the scope of ACP and presents novel data. However, the presentation of the numerical data is currently not up to the desired accuracy. Specifically, it is not always clear if presented percentage numbers are absolute or relative, and uncertainty values are mostly missing or not labeled sufficiently.

The used methods and assumptions are mostly clear and valid, yet some conclusions appear vague and a little too uncritical. I suggest to revise the manuscript and focus on accuracy and the specific comments given below.

Specific comments

Title: Use a more declarative title. A suggestion: **Fossil fuel combustion and biomass burning sources of global black carbon from GEOS-Chem for 2007-2013 compared to observations from 2002-2014**

$f_{bb}$ values: Is there a reason that no error values or standard deviations are given? In Table S1 all presented values include some form of error. I am writing "some form" because it is not clear what the error value in the table means statistically. What is the confidence interval?. Also, sometimes it remains unclear if the values refer to atmospheric or deposition values. For example in L11-17[1]. Please be more specific. Are the presented numbers for atmosphere or deposition?

- L12: Be more specific. What do you mean by "comparable contribution"? Give a number or range.
- L17: Be more specific. What do you mean with "reduction of discrepancies"? Discrepancies between what?
- L18: Be more specific. Which "discrepancies" do you mean?
- L23: Explain what the numbers in the brackets mean. What does this range refer to?
- L24-25: This sentence is not very clear. What does increase? Is it concentration, ratios, size or something else? Does "the former" refer to fossil fuels? Replace "the former" with a more specific description.
- L26-27: Is this finding novel? Please put this, your final conclusion of the abstract, in a bigger context.
- Abstract in general: Please write a sentence that explains your 4 experiments, in brief and put it before "We find" in L17.
* * *
[1] L refers to line numbers

- L14-16: You write: "Fossil fuel combustion often has an anthropogenic origin". When does it not have an anthropogenic origin? Please change this sentence.
- L23-25: The studies you refer to all looked at local emissions only. Besides that, their assumption is basically the conclusion of your study. Can you reflect more on this in the discussion part of your study?

- L5-15: The model description is very very short. The study would benefit from more detail. Explain why you picked this model and why you think that exactly this set-up (i.e. microphysics scheme, particle size resolution, model resolution, and emission inventories) is best suitable to run the four different tests you laid out before.
- L5-6: Please give a reference
- L11: Discuss in your discussion section what it means for your analysis if you compare observations from 2002-2014 to emission inventories from 2007 publications (in reality the emission information is older than that) or 2017 (in case of Asia).
- L12: Discuss here how you account for potential double counting of open fire emissions, included as agricultural burning in emission inventories and as open fires in GFED data.
- L15: briefly explain what updates you applied to the wet deposition scheme of Liu et al., 2001.
- L16-23: This part appears to be results, rather than model description. Please move it.
- L17: Explain why you only simulate 2007-2009 for Fig 1S.
- L19: What do you mean by "preferably" ? Can you quantify it?
- L24-25:
  - Please explain what the criteria was for picking the 41 sites. There is more data available for the observational period you picked (2002-2014) (e.g., (Zotter et al., 2014). You even cite at least one study, who's data you have not included in your analysis, despite falling in that time period (Winiger et al., 2017). Also, is it true that there are only data available from the Northern hemisphere? How confident can you be regarding simulations of the Southern hemisphere, if there is no observational data available?
  - Why are only the atmospheric samples mentioned in Table S1, when you also discuss the (snow) deposition measurements in your study?
  - Please indicate the measurement technique in Table S1, since there can be considerable differences in BC values between different observational methods (i.e. protocols).
  - Please indicate what the error values for $f_{bb}$ mean in Table S1.
  - Please fix the altitude values in Table S1. Some are 0, e.g., The station in Barrow is at 11 meters above sea level.
  - The reference for Szidat et al., 2004 (in Table S1) is missing in the SI references.
- L25-27: "Generally" is not really accurate. It depends largely on the season. There are cities (in Europe and Asia) where considerable (>40%) fbb values are observed in winter (Zotter et al., 2014, Bikkina et al., 2019)

- L9-L13: This seems to be an important point. The difference in the used methods is bigger than the uncertainties you found with your four experiments. Could you extend on this point? Additionally, add information of the used methods to Table S1.
- L10: I would be careful with highlighting single methods. Water extraction is also known to have considerable drawbacks (Azeem et al., 2017). A method for EC extraction (and BC measurements in general, actually) that satisfies all needs is currently nonexistent.
- L15: the factor 2 in Fig.1 (a) is not very clear. Some values (including error) go beyond the factor 2. Could you give these numbers in a table in the Supplementary Information (SI) ?
- L16-22: Please rewrite this. Within the context of these 3 sentences you write 3 times that "the low bias of $f_{bb}$ in [region X] is due to underestimation of biofuel combustion" in one form or another.
- L18: What do you mean by "current emission inventories"? Can you give references? You use an emission inventory with emissions older than 2007. Is that also a current emission inventory?
- L20: While there is little doubt that it is a great study, Qi et al., 2017c is hardly a reference for the sentence that precedes it. It is also not very specific what "large" means. Based on AMAP 2015 (see e.g., their Figure 11.1) European contribution to the BC burden in the Arctic is smaller than that of Russia or East and South Asia (and some other regions, based on the used model). Please consider rewriting this.
- L23: Where can we see the "factor of 2"? This is a general issue in the manuscript. These numbers ("factor of [X]") are presented in many places in the narrative but it is hard to follow. Would it be possible to provide a table in the SI?
- L23: Is there a reason you discuss possible reasons for underestimation of fbb in the Arctic, but decide to discuss overestimates of fbb in North America somewhere else, in Section 4.2.1.?
- L24-25: What makes you so certain that the large variations of fbb values is "due to the coarse horizontal and vertical resolutions."? Why shouldn't this be the case for all your other modelled cites as well? North-America, for example, has quite some orography as well.
- L30: You write that the model captures the spatial and temporal variations of fbb in BC deposition in this region [Himalayan-Tibetan plateau] and that "GEOS-Chem reproduces the average fbb in snow perfectly". How does that fit with the previous comment (L24-25)? Could you discuss this here, instead of section 4.2.1? Your analysis should also be more critical towards the model. What makes you think that the model simulates precipitation well, but less so atmospheric concentrations? Especially since many (if not all) other models struggle with orography (this GEOS-Chem run is at 4°x5° resolution and results discussed are in the Himalaya's !) and deposition schemes (e.g. Textor et al., 2006, Tegen et al., 2019).
- L31-L2 page 5: This sentence is good, because it gives the %-values that makes it possible to follow what you mean with "factor 2.3".
- fbb values in general: Please give and define uncertainties for your values or justify why you are not doing so.

- L7-9: You are using annual fossil fuel and biofuel emissions. Why would you expect anything else than "seasonal variation is relatively flat"? Or did you apply some sort of seasonality? This is not clear from reading the model description. Where (daily

resolved) open fires (from GFED) too weak in seasonality to show a difference? Or do you attribute this to model resolution entirely?

- L11: "probably underestimated" is a very vague formulation. Please be more accurate and specific.
- L12: It is unclear what you mean with "The similar magnitudes". Similar to what? Consequently, you conclusion of summer values (L12-14) " atmospheric fbb is largely determined by local emissions " is incomprehensible and appears speculative.
- L14: please specify "other seasons".
- L14: please specify if you mean model or observational "atmospheric fbb values".
- L9-15: This entire section appears vague and speculative. Please revise it.
- L15-18: Please give references and write that these are observational values.
- L24: " along the Mustang valley and Langtang valley. " To me, it is not clear where these valleys are located. Can you show a figure in the SI?
- L24-27: Would you expect any such local effects with the model resolution you are using? Is your model fit for purpose (coarse resolution and no annual resolved emissions) to compare seasonality's in such a terrain? Your conclusion (L29-30) seems not well supported, given the previously mentioned model limitations.
- L31: What do you mean with " no statistically significant differences "? Since you reference the figures here, you could write "no big differences". Better even, give a value for the variations: "no big differences (+/- XX %)".
- L32: which "four sites" do you mean?

- L4-5: What kind of solid fuel would be mostly used, i.e. what are the limited effects on fbb in the atmosphere? According to Mouteva et al., 2017, fraction modern (BC) doesn't change much throughout the year.
- L5-7: I don't think this sentence ("which is proved to have limited effects on fbb (Mouteva et al., 2017)") makes much sense. fbb does not vary much throughout the year. Why would one expect different values during strong winter inversions?
- L7-9: Which region are you referring to in this sentence? " Modeled fbb in the atmosphere is much higher than the fbb values of local emissions, suggesting a large regional effect on fbb in this region." I assume it is Salt Lake City, but it is not clear. And where do the numbers come from in " The model overestimates fbb in all seasons by a factor of 2–4. "
- L9-10: How does "This mismatch of model representation and observations partly explain[s] the large positive bias of fbb."? 'The model overestimates fbb' is not exactly an explanation of why we see this mismatch.
- L11-12: What kind of regional effect do you mean? What if emissions are wrong? This is an oversimplified analysis.
- L13-14: Which part in the "Comparison of fbb in local emissions and in the atmosphere" suggests that "fbb at MCOH is largely affected by long-range transport..."? Figure 2(g) does not even show if there are local emissions or not.
- L14-15: Please rewrite to reflect that both MOCH and SINH have relatively high fbb (~50%), very similar to each other.
- L16-30: Please state more clearly when you speak of observation or model.
- L23-26: It is unclear where the numbers here come from. Also, the highest fbb in the Northern hemisphere is found in African countries, according to your Fig. 3.
- L27-30: Two things: 1. It appears that Antarctica has strong seasonal fbb variation in atmosphere and deposition as well. However, it is a bit hard to quantify this from the

figure alone. 2. the "large contribution from open fire emissions" can not be seen in Figs S6 and S7, which show fbb alone. We would need to see BC concentrations before we can conclude that high biomass and biofuel emissions are responsible for the seasonality at the poles.

- L29-30: Two things: 1. To write that continental/regional (modelled) data is consistent with observations, when you have mostly one site per region (Fig 2) is a bit of a stretch. Please reflect this limitation more accurately or include more data, e.g., with a table in the SI. 2. Technically you do have one site for Europe, but there (Abisko) is a clear fbb seasonality.

- L3-8: I suggest moving this section to the introduction.
- L8: Could you give additional information why you chose to doubled biofuel emissions from domestic heating north of 45°N ? Why doubling? why domestic heating ? And why N of 45°N? There is, for example, evidence that open biomass burning might also be underestimated in the Arctic (Konovalov et al., 2018).
- L11-12: Could you reformulate this? It is not clear to me what you mean by "the model discrepancies... are reduced from -XY% to -YZ%"
- L20: What are the base run conditions for % of hydrophilic fresh biomass burning BC?
- L21-24: It is not clear what the percentages refer to exactly. If you say that the effect of this experiment lowers atmospheric fbb by "up to 11%", do you mean that the absolute fbb value decreases from e.g., 51% to 40% (-11%) or from 51% to ~46% (-11%) ? In the second sentence you write that the largest reductions are "-7%". How does 'largest reduction of -7%' stand in relation to 'lowering of up to 11%' ? Please revise these sentences and check the rest of the manuscript as well, where such comparisons take place. Again, a table would eliminate such uncertainties.
- L26: Can you quantify "large precipitation" ?
- L27-28: It is unclear what you are trying to say with this sentence.
- L29: Which region are you referring to?

- L25: What do the values in brackets mean?
- L29-30: Explain why you chose these diameters, since they diverge from the numbers you referred to in L20-21.
- L30: "Size resolved coagulation, condensation, nucleation and cloud processing" are implemented in TOMAS? Please specify or give a reference.
- L31-L1 page 9: That is an interesting find. Can you elaborate more on the reasons that lead to this effect (i.e. the larger decrease of fossil BC)?

- L20: "within 30%" is a bit unclear. Please be more specific.
- L28: increase " by 18–23%" is a bit unclear. Please be more specific. Essentially this is the same issue with my comment on L21-24, page 7 (see above).

- L8-12: This last paragraph is really disconnected from the rest of the text. Your research doesn't look into mixing states (internal vs external) and lensing. Consider changing it to reflect the scope better.

Figure 1
- Please use more descriptive axes, e.g., "modelled atmospheric BC fraction from biomass burning [%]"
- what standard deviation is shown in the figures?
- Could you show error bars in (b) ?
- consider different colors in (a) like in your previous publications (e.g. Qi et al., 2017)
- consider using different shapes in (b)

Figure 2
- Why do some sites not have emission bars?
- Please use more descriptive axes
- There is observational data for Barrow summers available now, see Winiger et al., 2019.
- Sometimes the standard deviation bars are not visible. Are they missing? Please specify
- Is the model grid for each site 4° x5°? Please specify

Figure 3
- Justify why large parts of Indonesia are missing in your regional analysis.

Figure 4
- Please use more descriptive axes
- include error bars if possible

Figure S2
- Why is the data for 2007-2009, unlike the rest of your study?

Colorscale
- Just a general comment. Consider a gradual color scale for your maps in future work, that doesn't contain rainbow colors. This is friendlier for color blinds and has the co-benefit that b/w prints are better understandable.

Technical corrections

Please consider the following suggestions:

- L12: remove: Specifically
- L17: replace "northern than" with "north of"
- L17-19: Redundancy. Remove "in winter" or "in cold season"
- L30: Quantify "large" or remove it.

- L2: I assume "BC distribution" refers to atmospheric surface concentrations? Do clarify.
- L5: Remove "In addition"
- L7: Remove "Moreover"
- L8-9: Remove the last sentence as it is redundant.

- L10: replace "separating" with "distinguishing".
- L16: insert "come" between "can" and "from"
- L22: There appears to be a word missing after "Alpine"
- L23: change "compare" to "compared"
- L24-25: change to: "The assumption behind these studies is, that the major..."
- L31: replace "following" with "consecutive"
- L31: remove "so far"
- L32: insert "in terms" between "sources" and "of global BC"

- L8: change "3 mn" to "3 nm"
- L9: What does MERRA2 stand for?
- L20: change " NC_Northest" to " NC_Northeast". Northeast is also misspelled twice in Fig S2 (legend inside right figure).

- L3-4: Please give a reference
- L4: change "end member" to "end members"
- L4-5: Please give a reference
- L8-9: Please give a reference
- L9: change "thermal-optical method" to " thermal-optical methods"
- L15: change "GEOS-Chem simulated fbb in the atmosphere agree with observations within a factor of 2" to "GEOS-Chem simulated atmospheric fbb agrees with observations within a factor of 2"
- L24: consider using "horizontal lines" instead of "error bars". The lines show a variation and not actual errors or deviations.

- L11: change " to site Abisko " to " to the Abisko site "
- L12: give a reference for the Barrow values.
- L23: "are shown" where?

- L3: "area sources" sounds a bit odd. Maybe "non-mobile" is better in this context. Or explain what you mean by it, like they do in Mouteva et al., 2017. (page 9851).
- L3: specify "solid" as "wood and coal".
- L10: Please start a new paragraph for Tokyo data.
- L16: consider changing to " 4.2.2 Spatial variation of modelled fbb"
- L22: change S3 to S4

- L9: In the first instance that "Exp." appears, please write what you mean by it.
- L10: change " Specifically " to " As a result "

- L25: specify $\kappa$
- L26-27: This sentence is redundant. Consider removing it

- change leaded to led
- Healy et al., 2015 appears to be missing as a reference.

References

AMAP Assessment 2015: Black carbon and ozone as Arctic climate forcers. Arctic Monitoring and Assessment Programme (AMAP), Oslo, Norway. vii + 116 pp.

H.A. Azeem, J. Martinsson, K.E. Stenström, E. Swietlicki, M. Sandahl, Towards the isolation and estimation of elemental carbon in atmospheric aerosols using supercritical fluid extraction and thermo-optical analysis, Anal. Bioanal. Chem., 409, pp. 4293-4300, 10.1007/s00216-017-0380-0, 2017

Konovalov et al., Estimation of black carbon emissions from Siberian fires using satellite observations of absorption and extinction optical depths, Atmos. Chem. Phys., 18, 14889-14924, https://doi.org/10.5194/acp-18-14889-2018, 2018

Bikkina et al., Air quality in megacity Delhi affected by countryside biomass burning, Nature Sustainability, 2, 200–205, https://doi.org/10.1038/s41893-019-0219-0, 2019

Tegen, I., Neubauer, D., Ferrachat, S., Siegenthaler-Le Drian, C., Bey, I., Schutgens, N., Stier, P., Watson-Parris, D., 10 Stanelle, T., Schmidt, H., Rast, S., Kokkola, H., Schultz, M., Schroeder, S., Daskalakis, N., Barthel, S., Heinold, B., and Lohmann, U.: The aerosol-climate model ECHAM6.3-HAM2.3: Aerosol evaluation, Geosci. Model Dev., 12, 1643-1677, https://doi.org/10.5194/gmd-12-1643-2019, 2019

Textor, C., Schulz, M., Guibert, S., Kinne, S., Balkanski, Y., Bauer,S., Berntsen, T., Berglen, T., Boucher, O., Chin, M., Dentener, F.,Diehl, T., Easter, R., Feichter, H., Fillmore, D., Ghan, S., Ginoux,P., Gong, S., Grini, A., Hendricks, J., Horowitz, L., Huang, P.,Isaksen, I., Iversen, I., Kloster, S., Koch, D., Kirkevåg, A., Krist-jansson, J. E., Krol, M., Lauer, A., Lamarque, J. F., Liu, X., Mon-tanaro, V., Myhre, G., Penner, J., Pitari, G., Reddy, S., Seland, Ø.,Stier, P., Takemura, T., and Tie, X.: Analysis and quantificationof the diversities of aerosol life cycles within AeroCom, Atmos.Chem. Phys., 6, 1777–1813, https://doi.org/10.5194/acp-6-1777-2006, 2006.

Qi, L., Li, Q., Li, Y., and He, C.: Factors controlling black carbon distribution in the Arctic, Atmospheric Chemistry and Physics, 17, 1037-1059, doi:10.5194/acp-17-1037-2017, 2017c.

Winiger, P., et al., Source apportionment of circum-Arctic atmospheric black carbon from isotopes and modeling, Sci. Adv. 5 (2), DOI: 10.1126/sciadv.aau8052, 2019

Zotter, P., Ciobanu, V. G., Zhang, Y. L., El-Haddad, I., Macchia, M., Daellenbach, K. R., Salazar, G. A., Huang, R.-J., Wacker, L., Hueglin, C., Piazzalunga, A., Fermo, P., Schwikowski, M., Baltensperger, U., Szidat, S., and Prévôt, A. S. H.: Radiocarbon analysis of elemental and organic carbon in Switzerland during winter-smog episodes from 2008 to 2012 – Part 1: Source apportionment and spatial variability, Atmos. Chem. Phys., 14, 13551-13570, https://doi.org/10.5194/acp-14-13551-2014, 2014.

---

## Referee Comment (RC2) · Anonymous Referee #1 · 5 May 2019

As one of the most important absorbing aerosols in the atmosphere, black carbon (BC) does play crucial roles in regional and global climate change. Both fossil fuel combustion and biomass burning contribute significantly to atmospheric BC, but its emission attributions are still not fully understood and are of great uncertainty. This work aims to quantify the contributions from different factors to sources of global BC in the atmosphere and in deposition by conducting global transport model and comparing it with the observations.The strength of this work is comprehensive observational data in multiple typical regions across the world.However, the authors jump to the conclusion several times in the interpretation of gaps between observational data and model results, and some bias are not clearly presented or fully investigated. Thus, more in-depth analysis ought to be provided. Here are some issues that need to be addressed

for further improving this work.

Section 2: The descriptions of simulation design and observations are far too simple. All the model configuration and simulations need to be introduced in detail. It is very confusing to understand EXP. A-D in Figure 4 without any introduction of these experiments in this part. And also, why these experiments are designed should be well documented.

As presented in Fig. 1-2, there does exist substantial gaps between model simulations and observational fbb in magnitude, seasonal variation, as well as spatial patterns. The authors generally describe the model bias and possible factors. However, more validation and detailed comparison may provide further in-depth information on model performance and uncertainties in related processes.

1. In addition to observed and GEOS-Chem simulated fraction of biomass burning of BC, information on the model performance on BC magnitude in different seasons may be helpful to understand the causes of the biases. Fig. S1 compared the Observed and GEOS-Chem simulated annual BC concentration but missed its seasonality and regional discrepancies. Since this work gathered carbon isotope analysis of BC at dozens of sites across the globe in different seasons, I do think detailed comparison and analysis on seasonal and regional bias of model simulation worth to be conducted.

2. As pointed out by the author, the bias in model results of fraction of biomass burning of BC can be greatly attributed to the lack of seasonality of existing fossil and biofuel combustion since that biomass burning emissions feature substantial temporal variations. To avoid the systematic bias caused by crude treatment of emission sources, monthly global emission inventory like EDGAR or HTAPv2 data or the monthly profile therein can be used as emission input of the model.

3. Since BC is one of typical primary pollutants in the atmosphere, transport process is of great importance besides emission sources. The relatively coarse spatial resolution (4° latitude × 5° longitude ) is not capable to capture some subtle meteorological conditions, which is vital for BC's transport and diffusion. Additionally, the coarse resolution make us to reconsider the representativeness of these observational sites, especially those near the complex terrain or mixed land cover. Applying fine spatial resolution in GEOS-Chem model may help reduce the bias of the model.

Minor issues: Page 6,Line 2 and Line 19-20: Please list the reference here. Did the authors get this conclusion based on emission inventories or existing publications. Anyhow, this statement should be supported by data or references.

Page 3,Line 8: change to "3 nm"

Page 10,Line 1: "leaded" should be "led"

―――――――――――――――――――

---

## Author Comment (AC1) · 9 Jul 2019

**Referee #1**

**General Comments:**

*"This manuscript represents an extensive approach to compare black carbon (BC) fractions of biomass burning from observations all across the northern hemisphere to GEOS-Chem simulations. The authors further conduct four sets of experiments to test uncertainties associated with the modelling of BC biofuel emissions, BC hygroscopicity, BC ageing, and BC size-resolved scavenging. The overall presentation, language, and figures are of high quality. The study is within the scope of ACP and presents novel data. However, the presentation of the numerical data is currently not up to the desired accuracy. Specifically, it is not always clear if presented percentage numbers are absolute or relative, and uncertainty values are mostly missing or not labeled sufficiently. The used methods and assumptions are mostly clear and valid, yet some conclusions appear vague and a little too uncritical. I suggest to revise the manuscript and focus on accuracy and the specific comments given below."*

**Specific comments:**

1. *"Title: Use a more declarative title. A suggestion: Fossil fuel combustion and biomass burning sources of global black carbon from GEOS-Chem for 2007-2013 compared to observations from 2002-2014."*

**Response**: Revised.

2. *"$f_{bb}$ values: Is there a reason that no error values or standard deviations are given? In Table S1 all presented values include some form of error. I am writing "some form" because it is not clear what the error value in the table means statistically. What is the confidence interval? Also, sometimes it remains unclear if the values refer to atmospheric or deposition values. For example in L11-171. Please be more specific. Are the presented numbers for atmosphere or deposition?"*

**Response**: Excellent point. We've added standard deviations in the abstract and added a note in table S1. Clarified $f_{bb}$ values.

3. *"L12: Be more specific. What do you mean by "comparable contribution"? Give a number or range."*

**Response**: Revised accordingly.

4. *"L17: Be more specific. What do you mean with "reduction of discrepancies"? Discrepancies between what?"*

**Response**: Revised.

5. *"L18: Be more specific. Which "discrepancies" do you mean?  "*

**Response**: Revised.

6. *"L23: Explain what the numbers in the brackets mean. What does this range refer to?"*

**Response**: Revised accordingly.

7. *"L24-25: This sentence is not very clear. What does increase? Is it concentration, ratios, size or something else? Does "the former" refer to fossil fuels? Replace "the former" with a more specific description.*

**Response**: Clarified.

8. *"L26-27: Is this finding novel? Please put this, your final conclusion of the abstract, in a bigger context."*

**Response**: Revised accordingly.

9. *"Abstract in general: Please write a sentence that explains your 4 experiments, in brief and put it before "We find" in L17."*

**Response**: Done.

10. *"L14-16: You write: "Fossil fuel combustion often has an anthropogenic origin". When does it not have an anthropogenic origin? Please change this sentence."*

**Response**: Done.

11. *"The studies you refer to all looked at local emissions only. Besides that, their assumption is basically the conclusion of your study. Can you reflect more on this in the discussion part of your study?"*

**Response**: Added discussion in Sect. 5.

12. *"L5-15: The model description is very very short. The study would benefit from more detail. Explain why you picked this model and why you think that exactly this set-up (i.e. microphysics scheme, particle size resolution, model resolution, and emission inventories) is best suitable to run the four different tests you laid out before.*

**Response**: We put the description of the standard simulation in this part and the set-up of the four experiments Sect. 4.3, where we discuss the uncertainties associated with model treatment of various atmospheric processes. The reasons for the set-ups are discussed in Sect. 4.3 as well. We decided to leave this structure as it was for a better flow of the discussion of uncertainties. Because some of the uncertainty experiment designs are

based on the results of the standard simulation.

13. *"L5-6: Please give a reference"*

**Response**: Revised.

14. *"Discuss in your discussion section what it means for your analysis if you compare observations from 2002-2014 to emission inventories from 2007 publications (in reality the emission information is older than that) or 2017 (in case of Asia)."*

**Response**: Acknowledged the uncertainties associated with the emission inventories used in Sect. 4.3.6.

15. *"Discuss here how you account for potential double counting of open fire emissions, included as agricultural burning in emission inventories and as open fires in GFED data."*

**Response**: Open fire emissions were accounted for in a separate biomass emissions inventory and were not included in the biofuel emissions. Thus, there is no potential double counting of open fires from the two inventories.

16. *"briefly explain what updates you applied to the wet deposition scheme of Liu et al., 2001."*

**Response**: Points well taken. Revised accordingly.

17. *"L16-23: This part appears to be results, rather than model description. Please move it."*

**Response**: We moved this part to the beginning of Sect. 4.

18. *"L17: Explain why you only simulate 2007-2009 for Fig 1S."*

**Response**: The data are for 2007-2013. Revised.

19. *"L19: What do you mean by "preferably" ? Can you quantify it?"*

**Response**: Quantified by correlation coefficient.

20. *"L24-25: Please explain what the criteria was for picking the 41 sites. There is more data available for the observational period you picked (2002-2014) (e.g., (Zotter et al., 2014). You even cite at least one study, who's data you have not included in your analysis, despite falling in that time period (Winiger et al., 2017). Also, is it true that there are only data available from the Northern hemisphere? How confident can you be regarding simulations of the Southern hemisphere, if there is no observational data available?"*

**Response**: Points well taken. We've tried our best to collect carbon isotope measurements published around the globe. We do not select sites. We've added data from Winiger et al. (2017), Zotter et al. (2014) and Bikkina et al. (2019) Table S1 and relavant analysis in the manuscript.

To our knowledge, there are no data available from the Southern Hemisphere to constrain the model results. Our analysis is based only on model results. This might produce some uncertainties. We acknowledge as such in Sect. 4.3.6.

21. *"L24-25: Why are only the atmospheric samples mentioned in Table S1, when you also discuss the (snow) deposition measurements in your study?"*

**Response**: We only have carbon isotope analysis of BC in snow over the Tibetan Plateau (Li et al., 2016). We've cited the study in the manuscript.

22. *"Please indicate the measurement technique in Table S1, since there can be considerable differences in BC values between different observational methods (i.e. protocols)."*

**Response**: Done.

23. *"Please indicate what the error values for $f_{bb}$ mean in Table S1."*

**Response**: Done.

24. *"Please fix the altitude values in Table S1. Some are 0, e.g., The station in Barrow is at 11 meters above sea level."*

**Response**: Done.

25. *"The reference for Szidat et al., 2004 (in Table S1) is missing in the SI references."*

**Response**: Added.

26. *"L25-27: "Generally" is not really accurate. It depends largely on the season. There are cities (in Europe and Asia) where considerable (>40%) $f_{bb}$ values are observed in winter (Zotter et al., 2014, Bikkina et al., 2019)"*

**Response**: Points well taken. Clarified. We also added carbon isotope observations from Zotter et al. (2014) and Bikkina et al. (2019) in Table S1 and relevant analysis.

27. *"L9-L13: This seems to be an important point. The difference in the used methods is bigger than the uncertainties you found with your four experiments. Could you extend on this point? Additionally, add information of the used methods to Table S1."*

**Response**: The uncertainties associated with observations are added to Sect. 4.3.6. Added used methods to Table S1.

28. *"L10: I would be careful with highlighting single methods. Water extraction is also known to have considerable drawbacks (Azeem et al., 2017). A method for EC extraction (and BC measurements in general, actually) that satisfies all needs is currently nonexistent."*

**Response**: Excellent point. Revised.

29. *"L15: the factor 2 in Fig.1 (a) is not very clear. Some values (including error) go beyond the factor 2. Could you give these numbers in a table in the Supplementary Information (SI) ?"*

**Response**: Done. See Table S2.

30. *"L16-22: Please rewrite this. Within the context of these 3 sentences you write 3 times that "the low bias of fbb in [region X] is due to underestimation of biofuel combustion" in one form or another."*

**Response**: Done.

31. *"L18: What do you mean by "current emission inventories"? Can you give references? You use an emission inventory with emissions older than 2007. Is that also a current emission inventory?"*

**Response**: Revised.

32. *"L20: While there is little doubt that it is a great study, Qi et al., 2017c is hardly a reference for the sentence that precedes it. It is also not very specific what "large" means. Based on AMAP 2015 (see e.g., their Figure 11.1) European contribution to the BC burden in the Arctic is smaller than that of Russia or East and South Asia (and some other regions, based on the used model). Please consider rewriting this."*

**Response**: Revised.

33. *"L23: Where can we see the "factor of 2"? This is a general issue in the manuscript. These numbers ("factor of [X]") are presented in many places in the narrative but it is hard to follow. Would it be possible to provide a table in the SI?"*

**Response**: Done.

34. *"L23: Is there a reason you discuss possible reasons for underestimation of $f_{bb}$ in the Arctic, but decide to discuss overestimates of $f_{bb}$ in North America somewhere else, in Section 4.2.1.?"*

**Response**: the overestimate of $f_{bb}$ in North America is season dependent, so we discuss it in Section 4.2.1, where seasonal variations are discussed.

35. *"L24-25: What makes you so certain that the large variations of fbb values is "due to the coarse horizontal and vertical resolutions."? Why shouldn't this be the case for all your other modelled cites as well? North-America, for example, has quite some orography as well."*

**Response**: Points well taken. Revised.

36. *"L30: You write that the model captures the spatial and temporal variations of $f_{bb}$ in BC deposition in this region [Himalayan-Tibetan plateau] and that "GEOS-Chem reproduces the average $f_{bb}$ in snow perfectly". How does that fit with the previous*

*comment (L24-25)? Could you discuss this here, instead of section 4.2.1? Your analysis should also be more critical towards the model. What makes you think that the model simulates precipitation well, but less so atmospheric concentrations? Especially since many (if not all) other models struggle with orography (this GEOS- Chem run is at 4°x5° resolution and results discussed are in the Himalaya's !) and deposition schemes (e.g. Textor et al., 2006, Tegen et al., 2019)."*

**Response**: Revised.

37. *"L31-L2 page 5: This sentence is good, because it gives the %-values that makes it possible to follow what you mean with "factor 2.3"."*

**Response**: Revised accordingly.

38. *"$f_{bb}$ values in general: Please give and define uncertainties for your values or justify why you are not doing so."*

**Response**: Point well taken. Revised accordingly.

39. *"L7-9: You are using annual fossil fuel and biofuel emissions. Why would you expect anything else than "seasonal variation is relatively flat"? Or did you apply some sort of seasonality? This is not clear from reading the model description. Where (daily resolved) open fires (from GFED) too weak in seasonality to show a difference? Or do you attribute this to model resolution entirely?"*

**Response**: Revised. The emissions are clarified in Sect. 3.

40. *"L11: "probably underestimated" is a very vague formulation. Please be more accurate and specific."*

**Response**: Deleted.

41. *"L12: It is unclear what you mean with "The similar magnitudes". Similar to what? Consequently, you conclusion of summer values (L12-14) " atmospheric $f_{bb}$ is largely determined by local emissions " is incomprehensible and appears speculative."*

**Response**: Revised.

42. *"L14: please specify "other seasons"."*

**Response**: Done.

43. *"L14: please specify if you mean model or observational "atmospheric $f_{bb}$ values"."*

**Response**: Done.

44. *"L9-15: This entire section appears vague and speculative. Please revise it."*

**Response**: Revised.

45. *"L15-18: Please give references and write that these are observational values."*

**Response**: Done.

46. *"L24: " along the Mustang valley and Langtang valley. " To me, it is not clear where these valleys are located. Can you show a figure in the SI?"*

**Response**: Please see Figures 1 and 2 in Li et al. (2016).

47. *"L24-27: Would you expect any such local effects with the model resolution you are using? Is your model fit for purpose (coarse resolution and no annual resolved emissions) to compare seasonality's in such a terrain? Your conclusion (L29-30) seems not well supported, given the previously mentioned model limitations."*

**Response**: Revised.

48. *"L31: What do you mean with " no statistically significant differences "? Since you reference the figures here, you could write "no big differences". Better even, give a value for the variations: "no big differences (+/- XX %)"."*

**Response**: Revised accordingly.

49. *"L32: which "four sites" do you mean?"*

**Response**: Clarified.

50. *"L4-5: What kind of solid fuel would be mostly used, i.e. what are the limited effects on $f_{bb}$ in the atmosphere? According to Mouteva et al., 2017, fraction modern (BC) doesn't change much throughout the year."*

**Response**: Revised.

51. *"L5-7: I don't think this sentence ("which is proved to have limited effects on $f_{bb}$ (Mouteva et al., 2017)") makes much sense. $f_{bb}$ does not vary much throughout the year. Why would one expect different values during strong winter inversions?"*

**Response**: Deleted.

52. *"L7-9: Which region are you referring to in this sentence? " Modeled $f_{bb}$ in the atmosphere is much higher than the $f_{bb}$ values of local emissions, suggesting a large regional effect on $f_{bb}$ in this region." I assume it is Salt Lake City, but it is not clear. And where do the numbers come from in " The model overestimates $f_{bb}$ in all seasons by a factor of 2–4. ""*

**Response**: Clarified.

53. *"L9-10: How does "This mismatch of model representation and observations partly explain[s] the large positive bias of $f_{bb}$."? 'The model overestimates $f_{bb}$' is not exactly an explanation of why we see this mismatch."*

**Response**: Revised.

54. *"L11-12: What kind of regional effect do you mean? What if emissions are wrong? This is an oversimplified analysis."*

**Response**: Revised.

55. *"L13-14: Which part in the "Comparison of $f_{bb}$ in local emissions and in the atmosphere" suggests that "$f_{bb}$ at MCOH is largely affected by long-range transport..."? Figure 2(g) does not even show if there are local emissions or not."*

**Response**: Revised.

56. *"L14-15: Please rewrite to reflect that both MOCH and SINH have relatively high $f_{bb}$ (~50%), very similar to each other."*

**Response**: Revised accordingly.

57. *"L16-30: Please state more clearly when you speak of observation or model."*

**Response**: Revised accordingly.

58. *"L23-26: It is unclear where the numbers here come from. Also, the highest $f_{bb}$ in the Northern hemisphere is found in African countries, according to your Fig. 3."*

**Response**: Revised.

59. *"L27-30: Two things: 1. It appears that Antarctica has strong seasonal $f_{bb}$ variation in atmosphere and deposition as well. However, it is a bit hard to quantify this from the figure alone. 2. the "large contribution from open fire emissions" can not be seen in Figs S6 and S7, which show $f_{bb}$ alone. We would need to see BC concentrations before we can conclude that high biomass and biofuel emissions are responsible for the seasonality at the poles."*

**Response**: 1. Revised. 2. Clarified.

60. *"L29-30: Two things: 1. To write that continental/regional (modelled) data is consistent with observations, when you have mostly one site per region (Fig 2) is a bit of a stretch. Please reflect this limitation more accurately or include more data, e.g., with a table in the SI. 2. Technically you do have one site for Europe, but there (Abisko) is a clear $f_{bb}$ seasonality."*

**Response**: 1. Points well taken. Revised. 2. Abisko locates inside the Arctic circle. We use "south Europe" instead of "Europe" in the manuscript.

61. *"L3-8: I suggest moving this section to the introduction."*

**Response**: This part explains why we do the uncertainty analysis associated with biofuel emissions. We think it's better to leave this part in sect. 4.3.1.

62. *"L8: Could you give additional information why you chose to doubled biofuel emissions from domestic heating north of 45°N? Why doubling? why domestic heating ? And why N of 45°N? There is, for example, evidence that open biomass burning might also be underestimated in the Arctic (Konovalov et al., 2018)."*

**Response**: Revised.

63. *"L11-12: Could you reformulate this? It is not clear to me what you mean by "the model discrepancies... are reduced from -XY% to -YZ%""*

**Response**: Revised.

64. *"L20: What are the base run conditions for % of hydrophilic fresh biomass burning BC?"*

**Response**: Revised.

65. *"L21-24: It is not clear what the percentages refer to exactly. If you say that the effect of this experiment lowers atmospheric $f_{bb}$ by "up to 11%", do you mean that the absolute $f_{bb}$ value decreases from e.g., 51% to 40% (-11%) or from 51% to ~46% (-11%)? In the second sentence you write that the largest reductions are "-7%". How does 'largest reduction of -7%' stand in relation to 'lowering of up to 11%'? Please revise these sentences and check the rest of the manuscript as well, where such comparisons take place. Again, a table would eliminate such uncertainties."*

**Response**: Clarified.

66. *"L26: Can you quantify "large precipitation" ?"*

**Response**: Done.

67. *"L27-28: It is unclear what you are trying to say with this sentence."*

**Response**: Revised.

68. *"L29: Which region are you referring to?"*

**Response**: Revised.

69. *"L25: What do the values in brackets mean?"*

**Response**: Clarified.

70. *"L29-30: Explain why you chose these diameters, since they diverge from the numbers you referred to in L20-21."*

**Response**: The observed diameters are examples from two studies with both size and coating thickness observations (Revised in the manuscript). The sizes in the model are based on various studies.

71. *"L30: " Size resolved coagulation, condensation, nucleation and cloud processing" are implemented in TOMAS? Please specify or give a reference."*

**Response**: Clarified.

72. *"L31-L1 page 9: That is an interesting find. Can you elaborate more on the reasons that lead to this effect (i.e. the larger decrease of fossil BC)?"*

**Response**: Revised.

73. *"L20: "within 30%" is a bit unclear. Please be more specific."*

**Response**: Revised.

74. *"L28: increase " by 18–23%" is a bit unclear. Please be more specific. Essentially this is the same issue with my comment on L21-24, page 7 (see above)."*

**Response**: Revised.

75. *"L8-12: This last paragraph is really disconnected from the rest of the text. Your research doesn't look into mixing states (internal vs external) and lensing. Consider changing it to reflect the scope better."*

**Response**: Revised.

Figure 1

76. *"Please use more descriptive axes, e.g., "modelled atmospheric BC fraction from biomass burning [%]"."*

**Response**: Done.

77. *"what standard deviation is shown in the figures?"*

**Response**: Clarified.

78. *"Could you show error bars in (b)?"*

**Response**: The standard deviations of observed $f_{bb}$ values are publicly unavailable. We leave it as it was.

79. *"consider different colors in (a) like in your previous publications (e.g. Qi et al., 2017)"*

**Response**: Done.

80. *"consider using different shapes in (b)"*

**Response**: Done.

Figure 2

81. *"Why do some sites not have emission bars?"*

**Response**: All sites have emission bars, but some are too small to be visible.

82. *"Please use more descriptive axes"*

**Response**: Done.

83. *"There is observational data for Barrow summers available now, see Winiger et al., 2019."*

**Response**: Added.

84. *"Sometimes the standard deviation bars are not visible. Are they missing? Please specify"*

**Response**: See response to question #81.

85. *"Is the model grid for each site 4° x5°? Please specify"*

**Response**: Done.

Figure 3

86. *"Justify why large parts of Indonesia are missing in your regional analysis."*

**Response**: Included Southeast Asia in the analysis.

Figure 4

87. *"Please use more descriptive axes"*

**Response**: Done.

88. *"include error bars if possible"*

**Response**: This is the relative change of mean $f_{bb}$ in each region ($r = ([BC]_{Exp.} - [BC]_{Std.})/[BC]_{Std}$). No error bars.

Figure S2

89.  *"Why is the data for 2007-2009, unlike the rest of your study?"*

**Response**: Revised.

Colorscale

90. *"Just a general comment. Consider a gradual color scale for your maps in future work, that doesn't contain rainbow colors. This is friendlier for color blinds and has the co- benefit that b/w prints are better understandable."*

**Response**: Sure. Thanks for your suggestions.

*Technical corrections*

*"Please consider the following suggestions:"*

91. *" L12: remove: Specifically"*

**Response**: Done.

92. *"L17: replace "northern than" with "north of""*

**Response**: Done.

93. *" L17-19: Redundancy. Remove "in winter" or "in cold season""*

**Response**: Done.

94. *" L30: Quantify "large" or remove it."*

**Response**: Removed.

95. *"L2: I assume "BC distribution" refers to atmospheric surface concentrations? Do clarify."*

**Response**: Clarified.

96. *" L5: Remove "In addition""*

**Response**: Done.

97. *" L7: Remove "Moreover""*

**Response**: Done.

98. *" L8-9: Remove the last sentence as it is redundant."*

**Response**: Done.

99. *" L10: replace "separating" with "distinguishing"."*

**Response**: Done.

100. *" L16: insert "come" between "can" and "from""*

**Response**: Done.

101. *L22: There appears to be a word missing after "Alpine""*

**Response**: Revised.

102. *L23: change "compare" to "compared""*

**Response**: Done.

103. *L24-25: change to: "The assumption behind these studies is, that the major...""*

**Response**: Done.

104. *L31: replace "following" with "consecutive""*

**Response**: Done.

105. *L31: remove "so far""*

**Response**: Done.

106. *L32: insert "in terms" between "sources" and "of global BC""*

**Response**: Done.

107. *L8: change "3 mn" to "3 nm""*

**Response**: Done.

108. *"L9: What does MERRA2 stand for?"*

**Response**: Clarified.

109. *"L20: change " NC_Northest" to " NC_Northeast". Northeast is also misspelled twice in Fig S2 (legend inside right figure)."*

**Response**: Done.

110. *" L3-4: Please give a reference"*

**Response**: Done.

111. *" L4: change "end member" to "end members""*

**Response**: Done.

112. *"L4-5: Please give a reference"*

**Response**: Done.

113. *"L8-9: Please give a reference"*

**Response**: Done.

114. *"L9: change "thermal-optical method" to " thermal-optical methods""*

**Response**: Done.

115. *"L15: change "GEOS-Chem simulated fbb in the atmosphere agree with observations within a factor of 2" to "GEOS-Chem simulated atmospheric fbb agrees with observations within a factor of 2""*

**Response**: Done.

116. *"L24: consider using "horizontal lines" instead of "error bars". The lines show a variation and not actual errors or deviations."*

**Response**: Points well taken. Revised.

117. *"L11: change " to site Abisko " to " to the Abisko site ""*

**Response**: The sentence is deleted.

118. *"L12: give a reference for the Barrow values."*

**Response**: Those are model results. Clarified in the manuscript.

119. *"L23: "are shown" where?"*

**Response**: Clarified.

120. *"L3: "area sources" sounds a bit odd. Maybe "non-mobile" is better in this context. Or explain what you mean by it, like they do in Mouteva et al., 2017. (page 9851)."*

**Response**: Revised accordingly.

121. *"L3: specify "solid" as "wood and coal"."*

**Response**: Revised.

122. *"L10: Please start a new paragraph for Tokyo data."*

**Response**: Done.

123. *"L16: consider changing to " 4.2.2 Spatial variation of modelled fbb""*

**Response**: Points well taken. Done.

124. *" L22: change S3 to S4"*

**Response**: Done.

125. *"L9: In the first instance that "Exp." appears, please write what you mean by it."*

**Response**: Done.

126. *"L10: change " Specifically " to " As a result ""*

**Response**: Done.

127. *"L25: specify k"*

**Response**: Done.

128. *"L26-27: This sentence is redundant. Consider removing it"*

**Response**: Done.

129. *"change leaded to led"*

**Response**: Done.

130. *" Healy et al., 2015 appears to be missing as a reference."*

**Response**: Revised.

**Fossil fuel combustion and biomass burning sources of global black carbon from GEOS-Chem and carbon isotope measurements**

Ling Qi[1] and Shuxiao Wang[1,2]

[1]State Key Joint Laboratory of Environment Simulation and Pollution Control, School of Environment, Tsinghua University, Beijing 100084, China
[2]State Environmental Protection Key Laboratory of Sources and Control of Air Pollution Complex, Beijing 100084, China

*Correspondence to*: Shuxiao Wang (shxwang@tsinghua.edu.cn)

**Abstract.** We identify sources (fossil fuel combustion versus biomass burning) of black carbon (BC) in the atmosphere and in deposition using a global 3D chemical transport model GEOS-Chem. We validate the simulated sources against carbon isotope measurements of BC around the globe and find that the model reproduces mean biomass burning contribution ($f_{bb}$, %) in various regions within a factor of 2 (except in Europe, where $f_{bb}$ is underestimated by 63%). GEOS-Chem shows that contribution from biomass burning in the Northern Hemisphere ($f_{bb}$: $35\pm14$%) is much less than that in the Southern Hemisphere ($50\pm11$%). The largest atmospheric $f_{bb}$ is in Africa ($64\pm20$%). Comparable contributions from biomass burning and fossil fuel combustion are found in South (S.) Asia ($53\pm10$%), Southeast (SE.) Asia ($53\pm11$%), S. America ($47\pm14$%), S. Pacific ($47\pm7$%), Australia ($53\pm14$%) and the Antarctic ($51\pm2$%). $
[revised manuscript text omitted]

| 6 | Arctic | Barrow | 71.2 | -156.6 | 11 | 2013 | Mar-May | spring | 18±7 | NIOSH 5040 | Winiger et al., 2019 |
| 7 | Arctic | Barrow | 71.2 | -156.6 | 11 | 2013 | Jul-Aug | summer | 34±5 | NIOSH 5040 | Winiger et al., 2019 |
| 8 | Arctic | Barrow | 71.2 | -156.6 | 11 | 2013 | Sep-Nov | fall | 9±5 | NIOSH 5040 | Winiger et al., 2019 |
| 9 | Arctic | Barrow | 71.2 | -156.6 | 11 | 2013 | Dec-Feb | winter | 34±9 | NIOSH 5040 | Winiger et al., 2019 |
| 10 | Arctic | Alert | 82.3 | -62.3 | 210 | 2014-15 | Feb | winter | 39±5 | NIOSH 5040 | Winiger et al., 2019 |
| 12 | Arctic | Alert | 82.3 | -62.3 | 210 | 2014-15 | Mar | spring | 39±5 | NIOSH 5040 | Winiger et al., 2019 |
| 13 | Arctic | Alert | 82.3 | -62.3 | 210 | 2014 | May | spring | 39±5 | NIOSH 5040 | Winiger et al., 2019 |
| 11 | Arctic | Alert | 82.3 | -62.3 | 210 | 2014 | Jul | summer | 37±5 | NIOSH 5040 | Winiger et al., 2019 |
| 14 | Arctic | Alert | 82.3 | -62.3 | 210 | 2014 | Nov | fall | 40±5 | NIOSH 5040 | Winiger et al., 2019 |
| 15 | Arctic | Alert | 82.3 | -62.3 | 210 | 2014 | Dec | winter | 44±5 | NIOSH 5040 | Winiger et al., 2019 |
| 16 | Arctic | Tiksi | 71.4 | 128.5 | 11 | 2012-14 | Mar-May | spring | 25±0.2 | NIOSH 5040 | Winiger et al., 2017 |
| 17 | Arctic | Tiksi | 71.4 | 128.5 | 11 | 2012-14 | Jun-Aug | summer | 45±0.1 | NIOSH 5040 | Winiger et al., 2017 |
| 18 | Arctic | Tiksi | 71.4 | 128.5 | 11 | 2012-14 | Sep-Nov | fall | 48±0.1 | NIOSH 5040 | Winiger et al., 2017 |
| 19 | Arctic | Tiksi | 71.4 | 128.5 | 11 | 2012-14 | Dec-Feb | winter | 28±0.2 | NIOSH 5040 | Winiger et al., 2017 |
| 20 | South Asia | MCOH[1] | 6.8 | 73.3 | 15 | 2006 | Jan_Mar | winter | 68±6 | NIOSH 5040 | Gustafsson et al., 2009 |
| 21 | South Asia | MCOH | 6.8 | 73.3 | 15 | 2008-09 | Dec_Mar | winter | 53±5 | NIOSH 5040 | Budhavant et al., 2015 |
| 22 | South Asia | MCOH | 6.8 | 73.3 | 15 | 2008-09 | Mar_Nov | summer | 53±11 | NIOSH 5040 | Budhavant et al., 2015 |
| 23 | South Asia | SINH[2] | 18.3 | 73.7 | 1450 | 2006 | Mar_Apr | spring | 46±8 | NIOSH 5040 | Gustafsson et al., 2009 |
| 24 | South Asia | SINH | 18.3 | 73.7 | 1450 | 2008-09 | Dec_Mar | winter | 56±3 | NIOSH 5040 | Budhavant et al., 2015 |
| 25 | South Asia | SINH | 18.3 | 73.7 | 1450 | 2008-09 | Mar_Nov | summer | 48±8 | NIOSH 5040 | Budhavant et al., 2015 |
| 26 | South Asia | Delhi | 28.5 | 77.2 | 300 | 2011 | Dec-Feb | witner | 39 | NIOSH 5040 | Bikkina et al., 2019 |
| 27 | South Asia | Delhi | 28.5 | 77.2 | 300 | 2011 | Mar-May | spring | 24 | NIOSH 5040 | Bikkina et al., 2019 |
| 28 | South Asia | Delhi | 28.5 | 77.2 | 300 | 2011 | Jun-Aug | summer | 17 | NIOSH 5040 | Bikkina et al., 2019 |
| 29 | South Asia | Delhi | 28.5 | 77.2 | 300 | 2011 | Sep-Nov | fall | 31 | NIOSH 5040 | Bikkina et al., 2019 |
| 30 | Europe | Göteborg | 57.7 | 11.9 | 20 | 2005 | Feb | winter | 12±4 | THEODORE[7] | Szidat et al., 2009 |
| 31 | Europe | Göteborg | 57.7 | 11.9 | 20 | 2006 | Jun | summer | 12±3 | THEODORE | Szidat et al., 2009 |
| 32 | Europe | Råö | 57.3 | 11.9 | 10 | 2005 | Feb | winter | 38±5 | THEODORE | Szidat et al., 2009 |
| 33 | Europe | Zurich | 47.3 | 8.5 | 410 | 2002 | Aug | summer | 8±1 | THEODORE | Szidat et al., 2004 |
| 34 | Europe | Zurich | 47.3 | 8.5 | 410 | 2003 | Feb | winter | 29±5 | THEODORE | Szidat et al., 2006 |

| | | | | | | | | | | | |
|---|---|---|---|---|---|---|---|---|---|---|---|
| 35 | Europe | Zurich | 47.3 | 8.5 | 410 | 2003 | Mar | spring | 15±5 | THEODORE | Szidat et al., 2006 |
| 36 | Europe | Zurich | 47.3 | 8.5 | 410 | 2006 | Jan | winter | 29±4 | THEODORE | Sandradewi et al., 2008a |
| 37 | Europe | Dübendorf | 47.4 | 8.6 | 440 | 2007 | Oct | fall | 36±3 | Swiss_4S[8] | Zhang et al., 2012 |
| 38 | Europe | Roveredo | 46.2 | 9.1 | 298 | 2005 | Jan | winter | 60±6 | THEODORE | Szidat et al., 2007 |
| 39 | Europe | Roveredo | 46.2 | 9.1 | 298 | 2005 | Mar | spring | 58±6 | THEODORE | Szidat et al., 2007 |
| 40 | Europe | Roveredo | 46.2 | 9.1 | 298 | 2005 | Dec | winter | 74±10 | THEODORE | Sandradewi et al., 2008b |
| 41 | Europe | Roveredo | 46.2 | 9.1 | 370 | 07/08- | Dec-Feb | winter | 46 | Swiss_4S | Zotter et al., 2014 |
| 42 | Europe | Moleno | 46.3 | 8.9 | 254 | 2005 | Feb | winter | 17±7 | THEODORE | Szidat et al., 2007 |
| 43 | Europe | Moleno | 46.3 | 8.99 | 305 | 07/08- | Dec-Feb | winter | 28 | Swiss_4S | Zotter et al., 2014 |
| 44 | Europe | Reiden | 47.2 | 7.9 | 457 | 2006 | Feb | winter | 30±4 | THEODORE | Sandradewi et al., 2008a |
| 45 | Europe | Reiden | 47.2 | 7.9 | 510 | 07/08- | Dec-Feb | winter | 34 | Swiss_4S | Zotter et al., 2014 |
| 46 | Europe | Massongex | 46.2 | 6.1 | 400 | 2006 | Nov | fall | 36±4 | THEODORE | Perron et al., 2010 |
| 47 | Europe | Massongex | 46.2 | 6.1 | 400 | 2006 | Dec | winter | 36±4 | THEODORE | Perron et al., 2010 |
| 48 | Europe | Massongex | 46.2 | 6.1 | 452 | 08/09-11/12 | Dec-Feb | winter | 54 | Swiss_4S | Zotter et al., 2014 |
| 49 | Europe | Saxon | 46.1 | 7.1 | 460 | 2006 | Dec | winter | 32±4 | THEODORE | Perron et al., 2010 |
| 50 | Europe | Sion | 46.2 | 7.3 | 505 | 2006 | Dec | winter | 20±3 | THEODORE | Perron et al., 2010 |
| 51 | Europe | Brigerbad | 46.3 | 7.9 | 650 | 2006 | Dec | winter | 31±4 | THEODORE | Perron et al., 2010 |
| 52 | Europe | Payerne | 46.8 | 6.9 | 456 | 2006 | Jan | winter | 60±4 | Swiss_4S | Zhang et al., 2012 |
| 53 | Europe | Payerne | 46.8 | 6.9 | 456 | 2006 | Jun | summer | 44±3 | Swiss_4S | Zhang et al., 2012 |
| 54 | Europe | Payerne | 46.8 | 6.9 | 539 | 07/08-11/12 | Dec-Feb | winter | 51 | Swiss_4S | Zotter et al., 2014 |
| 55 | Europe | Barcelona | 41.3 | 2.1 | 80 | 2009 | Mar | spring | 15±3 | adapted THEODORE | Minguillón et al., 2011 |
| 56 | Europe | Barcelona | 41.3 | 2.1 | 80 | 2009 | Jul | summer | 9±4 | adapted THEODORE | Minguillón et al., 2011 |
| 57 | Europe | Montseny | 41.8 | 2.3 | 720 | 2009 | Mar | spring | 37±4 | adapted THEODORE | Minguillón et al., 2011 |
| 58 | Europe | Montseny | 41.8 | 2.3 | 720 | 2009 | Jul | summer | 23±5 | adapted THEODORE | Minguillón et al., 2011 |
| 59 | Europe | Bern-Bollwerk | 46.9 | 7.6 | 506 | 08/09-12/13 | Dec-Feb | winter | 22 | Swiss_4S | Zotter et al., 2014 |
| 60 | Europe | Sissach-West | 47.5 | 7.8 | 410 | 07/08-11/12 | Dec-Feb | winter | 43 | Swiss_4S | Zotter et al., 2014 |
| 61 | Europe | St.Gallen-Rorschacherstrasse | 47.4 | 9.4 | 457 | 07/08-11/12 | Dec-Feb | winter | 38 | Swiss_4S | Zotter et al., 2014 |
| 62 | Europe | Vaduz-Austrasse | 47.1 | 9.5 | 706 | 07/08-11/12 | Dec-Feb | winter | 45 | Swiss_4S | Zotter et al., 2014 |
| 63 | Europe | Zôrich-Kaserne | 47.3 | 8.5 | 457 | 07/08- | Dec-Feb | winter | 41 | Swiss_4S | Zotter et al., 2014 |

| | | | | | | | | | | | |
|---|---|---|---|---|---|---|---|---|---|---|---|
| | | | | | | 11/12 | | | | | |
| 64 | Europe | Basel-St.Johann | 47.6 | 7.6 | 308 | 07/08-08/09 | Dec-Feb | winter | 41 | Swiss_4S | Zotter et al., 2014 |
| 65 | Europe | Solothurn-Altwyberhôsli | 47.1 | 7.6 | 502 | 07/08-11/12 | Dec-Feb | winter | 46 | Swiss_4S | Zotter et al., 2014 |
| 66 | Europe | Sch_chental | 46.8 | 8.8 | 995 | 10/11- | Dec-Feb | winter | 67 | Swiss_4S | Zotter et al., 2014 |
| 67 | Europe | Chiasso | 45.8 | 9 | 291 | 07/08-11/12 | Dec-Feb | winter | 41 | Swiss_4S | Zotter et al., 2014 |
| 68 | Europe | Magadino-Cadenazzo | 46.8 | 6.9 | 254 | 07/08-11/12 | Dec-Feb | winter | 48 | Swiss_4S | Zotter et al., 2014 |
| 69 | Europe | San-Vittore | 46.2 | 9.1 | 330 | 07/08-11/12 | Dec-Feb | winter | 66 | Swiss_4S | Zotter et al., 2014 |
| 70 | North America | Salt Lake City | 40.7 | -111.8 | 1426 | 2012-14 | annual | summer | 11±1.1 | adapted Swiss_4S | Mouteva et al., 2017 |
| 71 | North America | Mexico City | 19.5 | -99.1 | 2240 | 2006 | Mar | spring | 16±4 | THEODORE | Aiken et al., 2010 |
| 72 | East Asia | Tokyo | 35.6 | 139.6 | 40 | 2004 | Oct | fall | 36.4 | adapted IMPROVE[9] | Yamamoto et al., 2007 |
| 73 | East Asia | Tokyo | 35.6 | 139.6 | 40 | 2004 | Dec | winter | 33.8 | adapted IMPROVE | Yamamoto et al., 2007 |
| 74 | East Asia | Tokyo | 35.6 | 139.6 | 40 | 2004 | Feb | winter | 32.6 | adapted IMPROVE | Yamamoto et al., 2007 |
| 75 | East Asia | Tokyo | 35.6 | 139.6 | 40 | 2004 | Apr | spring | 41.3 | adapted IMPROVE | Yamamoto et al., 2007 |
| 76 | East Asia | Tokyo | 35.6 | 139.6 | 40 | 2004 | Jun | summer | 37.7 | adapted IMPROVE | Yamamoto et al., 2007 |
| 77 | East Asia | Tokyo | 35.6 | 139.6 | 40 | 2004 | Aug | summer | 35.8 | adapted IMPROVE | Yamamoto et al., 2007 |
| 79 | East Asia | Beijing | 39.9 | 116.4 | 55 | 2013 | Jan | winter | 30±2 | Swiss_4S | Zhang et al., 2015 |
| 80 | East Asia | Beijing | 39.9 | 116.4 | 55 | 2013 | Jan | winter | 26±2 | NIOSH 5040 | Andersson et al., 2015 |
| 81 | East Asia | Beijing | 39.9 | 116.4 | 55 | 2010 | Feb | winter | 17±4 | NIOSH 5040 | Chen et al., 2013 |
| 82 | East Asia | Shanghai | 31.3 | 121.5 | 4 | 2013 | Jan | winter | 21±2 | Swiss_4S | Zhang et al., 2015 |
| 83 | East Asia | Shanghai | 31.3 | 121.5 | 4 | 2013 | Jan | winter | 32±2 | NIOSH 5040 | Andersson et al., 2015 |
| 84 | East Asia | Shanghai | 31.3 | 121.5 | 4 | 2010 | Jan | winter | 17±4 | NIOSH 5040 | Chen et al., 2013 |
| 85 | East Asia | Guangzhou | 23.1 | 113.4 | 15 | 2013 | Jan | winter | 48±5 | Swiss_4S | Zhang et al., 2015 |
| 86 | East Asia | Guangzhou | 23.1 | 113.4 | 15 | 2013 | Jan | winter | 32±2 | NIOSH 5040 | Andersson et al., 2015 |
| 89 | East Asia | Xi'an | 34.2 | 108.9 | 416 | 2013 | Jan | winter | 25±3 | | Zhang et al., 2015 |
| 90 | East Asia | Xiamen | 24.5 | 118 | 2 | 2009 | Dec | winter | 13±3 | NIOSH 5040 | Chen et al., 2013 |
| 93 | East Asia | KCOG[3] | 33.3 | 126.2 | 72 | 2011 | Mar | winter | 25±6 | NIOSH 5040 | Chen et al., 2013 |
| 94 | East Asia | SCCO[4] | 24.6 | 118.1 | 3 | 2009 | Jan | winter | 22±3 | NIOSH 5040 | Chen et al., 2013 |
| 95 | Tibet | Jilong | 28.2 | 86 | 4166 | 2013 | Apr | spring | 45 | NIOSH 5040 | Li et al., 2016 |
| 96 | Tibet | Jilong | 28.2 | 86 | 4166 | 2013 | Jun | winter | 41 | NIOSH 5040 | Li et al., 2016 |
| 97 | Tibet | Nielamu | 28.2 | 86 | 4166 | 2013 | Nov | fall | 40 | NIOSH 5040 | Li et al., 2016 |
| 98 | Tibet | Dhunche | 28.1 | 85.3 | 2051 | 2014 | Jan | winter | 49 | NIOSH 5040 | Li et al., 2016 |
| 99 | Tibet | Dhunche | 28.1 | 85.3 | 2051 | 2013 | Aug | summer | 16 | NIOSH 5040 | Li et al., 2016 |

| 100 | Tibet | Dhunche | 28.1 | 85.3 | 2051 | 2013 | Sep | fall | 41 | NIOSH 5040 | Li et al., 2016 |
|-----|-------|---------|------|------|------|------|-----|------|-----|------------|-----------------|
| 101 | Tibet | Bode | 27.7 | 85.4 | 1386 | 2014 | Jan | winter | 42 | NIOSH 5040 | Li et al., 2016 |
| 102 | Tibet | Bode | 27.7 | 85.4 | 1386 | 2013 | Apr | spring | 33 | NIOSH 5040 | Li et al., 2016 |
| 103 | Tibet | Bode | 27.7 | 85.4 | 1386 | 2013 | Aug | summer | 16 | NIOSH 5040 | Li et al., 2016 |
| 104 | Tibet | Bode | 27.7 | 85.4 | 1386 | 2013 | Nov | fall | 28 | NIOSH 5040 | Li et al., 2016 |
| 105 | Tibet | Zhongba | 29.7 | 84 | 4704 | 2013 | Apr | spring | 70 | NIOSH 5040 | Li et al., 2016 |
| 106 | Tibet | Jomsom | 28.8 | 83.7 | 3048 | 2013 | Apr | spring | 57 | NIOSH 5040 | Li et al., 2016 |
| 107 | Tibet | Pokhara | 28.2 | 84 | 813 | 2013 | Jul | summer | 26 | NIOSH 5040 | Li et al., 2016 |
| 108 | Tibet | Pokhara | 28.2 | 84 | 813 | 2013 | Apr | spring | 65 | NIOSH 5040 | Li et al., 2016 |
| 109 | Tibet | Lumbini | 27.5 | 83.3 | 100 | 2013 | Apr | spring | 58 | NIOSH 5040 | Li et al., 2016 |
| 110 | Tibet | Lumbini | 27.5 | 83.3 | 100 | 2013 | Jul | summer | 42 | NIOSH 5040 | Li et al., 2016 |
| 111 | Tibet | Lumbini | 27.5 | 83.3 | 100 | 2013 | Oct | fall | 53 | NIOSH 5040 | Li et al., 2016 |
| 112 | Tibet | Lumbini | 27.5 | 83.3 | 100 | 2013 | Dec | winter | 49 | NIOSH 5040 | Li et al., 2016 |
| 113 | Tibet | Namco | 30.8 | 91 | 4730 | 2013 | Apr | spring | 54 | NIOSH 5040 | Li et al., 2016 |
| 114 | Tibet | Namco | 30.8 | 91 | 4730 | 2014 | Jun | summer | 63 | NIOSH 5040 | Li et al., 2016 |
| 115 | Tibet | Namco | 30.8 | 91 | 4730 | 2014 | Jul | summer | 49 | NIOSH 5040 | Li et al., 2016 |
| 116 | Tibet | Namco | 30.8 | 91 | 4730 | 2013 | Nov | fall | 58 | NIOSH 5040 | Li et al., 2016 |
| 117 | Tibet | Lulang | 29.8 | 94.7 | 3326 | 2014 | Jun | summer | 20 | NIOSH 5040 | Li et al., 2016 |
| 118 | Tibet | Lulang | 29.8 | 94.7 | 3326 | 2014 | Jul | summer | 23 | NIOSH 5040 | Li et al., 2016 |
| 119 | Tibet | Lhasa | 29.6 | 91 | 3640 | 2014 | Jan | winter | 18 | NIOSH 5040 | Li et al., 2016 |
| 120 | Tibet | Lhasa | 29.6 | 91 | 3640 | 2013 | Apr | spring | 24 | NIOSH 5040 | Li et al., 2016 |
| 121 | Tibet | Lhasa | 29.6 | 91 | 3640 | 2013 | Jun | summer | 7 | NIOSH 5040 | Li et al., 2016 |

[1] Maldives Climate Observatory in Hanimaadhoo

[2] Indian Institute of Tropical Meteorology in Sinhagad, India

[3] Korea Climate Observatory-Gosan

[4] South China Climate Observatory

[5] Standard deviation of observations

[6] National Institute for Occupational Safety and Health 5040

[7] Two-step Heating system for the EC/OC Determination of Radiocarbon in the Environment

[8] four-step (S1, S2, S3 and S4) thermal-optical protocol

[9] Integragency Monitoring of Protected Visual Environments

Table S2 Observed and GEOS-Chem simulated atmospheric $f_{bb}$ in various regions (%)

| Region | Observations | Simulation |
|---|---|---|
| The Himalayan-Tibetan plateau | 39±17* | 62±7 |
| South Asia | 37±16 | 55±5 |
| East Asia | 29±9 | 31±5 |
| The Arctic | 33±14 | 32±23 |
| North America | 14±4 | 29±2 |
| Europe | 43±16 | 14±3 |

*Standard deviation, reflecting variations of atmospheric $f_{bb}$ among different sites during different seasons in each region.

[Figure]

**Figure S1**. (a) Probability density function of observed (red line) and GEOS-Chem simulated (black) BC concentrations in surface air (µg m$^{-3}$) and (b) Observed and GEOS-Chem simulated annual BC concentrations in surface air. Data are for 2007–2013. Solid line is 1:1 ratio line and dashed lines are 1:2 (or 2:1).

[Figure]

**Figure S2**. (a) Probability density function of observed (red line) and GEOS-Chem simulated (black) BC concentration in snow (ng g$^{-1}$) and (b) medians of observed and simulated BC in snow (ng g$^{-1}$) in the Arctic, North America (Canada, the Great Plains, the Pacific Northwest, and the Rockies, as defined in Doherty et al., 2014)), Northern China (Inner Mongolia, Northeast Border and Northeast Industrial, as defined by Wang et al., 2013), and Xinjiang, China. The regions are symbol-coded. Solid line – 1:1 ratio line; dashed lines – 1:2 (or 2:1) ratio lines.

[Figure]

**Figure S3**. Carbon isotope measurement stations of BC as listed in Table S1.

[Figure]

**Figure S4**. Average $f_{bb}$ of BC in surface atmosphere during March–May (MAM), June–August (JJA), September–November (SON) and December–February (DJF) for 2007–2013.

[Figure]

**Figure S5**. Same as Figure S4, but for BC deposition.

[Figure]

**Figure S6**. Same as Figure S4, but for BC emissions.

[Figure]

**Figure S7**. Average contribution of open burning to BC emissions (%) during March–May (MAM), June–August (JJA), September–November (SON) and December–February (DJF) for 2007–2013.

[Figure]

**Figure S8**. Observed and GEOS-Chem simulated mean $f_{bb}$ (%) (a) of BC in the atmosphere in the six regions in Northern Hemisphere and (b) of BC deposited in snow over the Tibetan plateau. The regions are symbol-coded and the simulations are color-coded (see text for details). Solid lines are 1:1 and dashed lines are 1:2 (or 2:1).

---

## Author Comment (AC2) · 9 Jul 2019

**Referee #2**

*Major Comments:*

*"As one of the most important absorbing aerosols in the atmosphere, black carbon (BC) does play crucial roles in regional and global climate change. Both fossil fuel combustion and biomass burning contribute significantly to atmospheric BC, but its emission attributions are still not fully understood and are of great uncertainty. This work aims to quantify the contributions from different factors to sources of global BC in the atmosphere and in deposition by conducting global transport model and comparing it with the observations. The strength of this work is comprehensive observational data in multiple typical regions across the world. However, the authors jump to the conclusion several times in the interpretation of gaps between observational data and model results, and some bias are not clearly presented or fully investigated. Thus, more in- depth analysis ought to be provided. Here are some issues that need to be addressed for further improving this work."*

1 *"Section 2: The descriptions of simulation design and observations are far too simple. All the model configuration and simulations need to be introduced in detail. It is very confusing to understand EXP. A-D in Figure 4 without any introduction of these experiments in this part. And also, why these experiments are designed should be well documented."*

**Response**: Why we design the experiments are documented in Sect. 4.3. Since the some of the uncertainty experiments are based on the results of the standard simulation, it's hard to move the description of Exps. A-D to the model description part. We decided to leave the exp. description in Sect. 4.3.

2 *"As presented in Fig. 1-2, there does exist substantial gaps between model simulations and observational fbb in magnitude, seasonal variation, as well as spatial patterns. The authors generally describe the model bias and possible factors. However, more validation and detailed comparison may provide further in-depth information on model performance and uncertainties in related processes.*

2.1. *In addition to observed and GEOS-Chem simulated fraction of biomass burning of BC, information on the model performance on BC magnitude in different seasons may be helpful to understand the causes of the biases. Fig. S1 compared the Observed and GEOS-Chem simulated annual BC concentration but missed its seasonality and regional discrepancies. Since this work gathered carbon isotope analysis of BC at dozens of sites across the globe in different seasons, I do think detailed comparison and analysis on seasonal and regional bias of model simulation worth to be conducted."*

**Response**: Seasonal variations of BC concentration in different regions do help understand the model bias of BC concentration, but provide limited information on bias of BC sources, since BC concentration and $f_{bb}$ have distinctively different seasonal

variations as we discussed in the manuscript in P6 L9-12. In addition, most of the seasonal observations of BC are publicly unavailable. We only have seasonal variations of BC from IMPROVE network in the United States (as discussed in Qi et al. (2017)) and in the Arctic. Sources of BC in the Arctic are discussed in detail in a separate manuscript (Qi and Wang, 2019 in review). Considering the limited sites (only two) with carbon isotope measurements, we think analyzing seasonal variations of BC concentration from IMPROVE measurements do not provide proper information for the source apportionment analysis.

**2.2** *"As pointed out by the author, the bias in model results of fraction of biomass burning of BC can be greatly attributed to the lack of seasonality of existing fossil and biofuel combustion since that biomass burning emissions feature substantial temporal variations. To avoid the systematic bias caused by crude treatment of emission sources, monthly global emission inventory like EDGAR or HTAPv2 data or the monthly profile therein can be used as emission input of the model. "*

**Response**: We apply seasonal variations to domestic heating (Sect. 2). Other sectors, such as industry and transport, have little or no seasonal variations. We use daily emissions of GFED4 for open fire emissions (Sect. 2). We also revised some of the related analysis in Sect. 4.

**2.3** *"Since BC is one of typical primary pollutants in the atmosphere, transport process is of great importance besides emission sources. The relatively coarse spatial resolution (4◦ latitude × 5◦ longitude) is not capable to capture some subtle meteorological conditions, which is vital for BC's transport and diffusion. Additionally, the coarse resolution make us to reconsider the representativeness of these observational sites, especially those near the complex terrain or mixed land cover. Applying fine spatial resolution in GEOS-Chem model may help reduce the bias of the model."*

**Response**: We added uncertainty analysis of model resolution in Sect. 4.3.5.

**Specific comments:**

**3**. *"Page 6, Line 2 and Line 19-20: Please list the reference here. Did the authors get this conclusion based on emission inventories or existing publications. Anyhow, this statement should be supported by data or references. "*

**Response**: Clarified.

**4**. *"Page 3,Line 8: change to "3 nm" "*

**Response**: Done.

**5**. *"Page 10,Line 1: "leaded" should be "led""*

**Response**: Done.

**References**

Qi, L., Li, Q., He, C., Wang, X., and Huang, J.: Effects of the Wegener–Bergeron–Findeisen process on global black carbon distribution, Atmospheric Chemistry and Physics, 17, 7459-7479, doi:10.5194/acp-17-7459-2017, 2017a.

Qi, L., and Wang, S.X.: Sources of black carbon in the atmosphere and in snow in the Arctic, Science of the Total Environment, 2019, in review.

[revised manuscript text omitted]

[1] Maldives Climate Observatory in Hanimaadhoo

[2] Indian Institute of Tropical Meteorology in Sinhagad, India

[3] Korea Climate Observatory-Gosan

[4] South China Climate Observatory

[5] Standard deviation of observations

[6] National Institute for Occupational Safety and Health 5040

[7] Two-step Heating system for the EC/OC Determination of Radiocarbon in the Environment

[8] four-step (S1, S2, S3 and S4) thermal-optical protocol

[9] Integragency Monitoring of Protected Visual Environments

Table S2 Observed and GEOS-Chem simulated atmospheric $f_{bb}$ in various regions (%)

| Region | Observations | Simulation |
| --- | --- | --- |
| The Himalayan-Tibetan plateau | 39±17* | 62±7 |
| South Asia | 37±16 | 55±5 |
| East Asia | 29±9 | 31±5 |
| The Arctic | 33±14 | 32±23 |
| North America | 14±4 | 29±2 |
| Europe | 43±16 | 14±3 |

*Standard deviation, reflecting variations of atmospheric $f_{bb}$ among different sites during different seasons in each region.

[Figure]

**Figure S1**. (a) Probability density function of observed (red line) and GEOS-Chem simulated (black) BC concentrations in surface air (μg m⁻³) and (b) Observed and GEOS-Chem simulated annual BC concentrations in surface air. Data are for 2007–2013. Solid line is 1:1 ratio line and dashed lines are 1:2 (or 2:1).

[Figure]

**Figure S2**. (a) Probability density function of observed (red line) and GEOS-Chem simulated (black) BC concentration in snow (ng g$^{-1}$) and (b) medians of observed and simulated BC in snow (ng g$^{-1}$) in the Arctic, North America (Canada, the Great Plains, the Pacific Northwest, and the Rockies, as defined in Doherty et al., 2014)), Northern China (Inner Mongolia, Northeast Border and Northeast Industrial, as defined by Wang et al., 2013), and Xinjiang, China. The regions are symbol-coded. Solid line – 1:1 ratio line; dashed lines – 1:2 (or 2:1) ratio lines.

[Figure]

**Figure S3**. Carbon isotope measurement stations of BC as listed in Table S1.

[Figure]

**Figure S4**. Average $f_{bb}$ of BC in surface atmosphere during March–May (MAM), June–August (JJA), September–November (SON) and December–February (DJF) for 2007–2013.

[Figure]

**Figure S5**. Same as Figure S4, but for BC deposition.

[Figure]

**Figure S6**. Same as Figure S4, but for BC emissions.

[Figure]

**Figure S7**. Average contribution of open burning to BC emissions (%) during March–May (MAM), June–August (JJA), September–November (SON) and December–February (DJF) for 2007–2013.

[Figure]

**Figure S8**. Observed and GEOS-Chem simulated mean $f_{bb}$ (%) (a) of BC in the atmosphere in the six regions in Northern Hemisphere and (b) of BC deposited in snow over the Tibetan plateau. The regions are symbol-coded and the simulations are color-coded (see text for details). Solid lines are 1:1 and dashed lines are 1:2 (or 2:1).